# Sand flies: Basic information on the vectors of leishmaniasis and their interactions with *Leishmania* parasites

Pedro Cecílio [1,2,3,4✉], Anabela Cordeiro-da-Silva[2,3,4] & Fabiano Oliveira [1✉]

Blood-sucking arthropods transmit a variety of human pathogens acting as disseminators of the so-called vector-borne diseases. Leishmaniasis is a spectrum of diseases caused by different *Leishmania* species, transmitted *quasi* worldwide by sand flies. However, whereas many laboratories focus on the disease(s) and etiological agents, considerably less study the respective vectors. In fact, information on sand flies is neither abundant nor easy to find; aspects including basic biology, ecology, and sand-fly-*Leishmania* interactions are usually reported separately. Here, we compile elemental information on sand flies, in the context of leishmaniasis. We discuss the biology, distribution, and life cycle, the blood-feeding process, and the *Leishmania*-sand fly interactions that govern parasite transmission. Additionally, we highlight some outstanding questions that need to be answered for the complete understanding of parasite–vector–host interactions in leishmaniasis.

Estimates point to the existence of 200 million insects alive per each human at any given point; among them, around 14,000 species feed on blood[1], some, with potentially severe implications for human health. In fact, diseases associated with arthropod vectors (generally known as vector-borne diseases) account for more than 17% of all infectious diseases, and cause at least 700,000 deaths annually, as per the most recent estimates[2,3]. Of note, since most of these diseases disproportionally affect individuals in resource-poor countries of the tropics and subtropics, they are considered Neglected Tropical Diseases (NTDs)[3,4]. In line with this notion, vector research has focused disproportionally on a few species (mostly mosquitoes associated with malaria and other diseases), and overlooked other arthropods still associated with a fairly high disease burden.

Among the abovementioned NTDs, leishmaniasis is associated with significant incidence, morbidity, and mortality (the deadliest NTD, according to recent global estimates)[5,6]. Leishmaniasis is a spectrum of diseases caused by around 20 *Leishmania* species, transmitted by different phlebotomine sand fly species (Table 1). Of note, many peculiarities of leishmaniasis were highlighted through the years, some of them, still without a clear explanation. For instance, while different *Leishmania* species are associated with similar clinical manifestations[7–9], not infrequently, the same parasite species is linked to distinct clinical pictures[10–12], suggesting that parasite tropism and/or virulence may not be the only pathogenesis determinants. This is supported by the many times repeated statement saying that infection and disease progression depends on "complex interactions between the parasite and the host's immune response"[13]. Still, while the "atypical leishmaniasis" presentations described in the context of immunocompromised individuals (e.g., malnourished, HIV positive, and pharmacologically immunosuppressed)

---

[1] Vector Molecular Biology Section, Laboratory of Malaria and Vector Research, National Institute of Allergy and Infectious Diseases, National Institutes of Health, Rockville, MD, USA. [2] i3S - Instituto de Investigação e Inovação em Saúde, Universidade do Porto, Porto, Portugal. [3] Parasite Disease Group, IBMC - Instituto de Biologia Molecular e Celular, Universidade do Porto, Porto, Portugal. [4] Departamento de Ciências Biológicas, Faculdade de Farmácia da Universidade do Porto (FFUP), Porto, Portugal. ✉email: pedro.amadocecilio@nih.gov; loliveira@niaid.nih.gov

**Table 1 Leishmaniasis: etiological, clinical, and epidemiological aspects, including the most relevant incriminated vectors.**

| Leishmania species | Clinical form | Main clinical features | Natural progression | Risk groups | Main reservoir | Transmission | Main vectors& | OW vs. NW | High-burden countries or regions | Estimated annual worldwide incidence |
|---|---|---|---|---|---|---|---|---|---|---|
| Leishmania (Leishmania) donovani | VL and PKDL | Persistent fever, splenomegaly, weight loss, and anemia in VL; multiple painless macular, papular, or nodular lesions in PKDL | VL is fatal within 2 years if untreated; PKDL lesions self-heal in up to 85% of cases in Africa but rarely in Asia | Predominantly adolescents and young adults for VL; young children in Sudan and no clearly established risk factors for PKDL | Humans | Epidemic anthroponotic | Phlebotomus argentipes, Phlebotomus orientalis, Phlebotomus martini, Phlebotomus alexandri | OW | India, Bangladesh, Ethiopia, Sudan, and South Sudan | 50,000–90,000 VL cases; unknown number of PKDL cases |
| Leishmania (Leishmania) tropica | CL, LR, and rarely VL | Ulcerating dry lesions, painless, and frequently multiple | CL lesions often self-heal within 1 year | No well-defined risk groups | Humans, hyraxes | Urban anthroponotic | Phlebotomus sergenti, Phlebotomus arabicus | OW | Eastern Mediterranean, the Middle East, and northeastern and southern Africa | 200,000–400 000 CL |
| Leishmania (Leishmania) aethiopica | CL, DCL, DsCL, and oronasal CL | Localized cutaneous nodular lesions; occasionally oronasal; rarely ulcerates | Self-healing, except for DCL, within 2–5 years | Limited evidence; adolescents | Hyraxes | Rural zoonotic | Phlebotomus longipes, Phlebotomus pedifer, Phlebotomus sergenti | OW | Ethiopia and Kenya | 20,000–40,000 CL |
| Leishmania (Leishmania) major | CL | Rapid necrosis, multiple wet sores, and severe inflammation | Self-healing in >50% of cases within 2–8 months; multiple lesions slow to heal, and severe scarring | No well-defined risk groups | Rodents | Rural zoonotic | Phlebotomus papatasi, Phlebotomus duboscqi, Phlebotomus salehi, Phlebotomus caucasicus | OW | Iran, Saudi Arabia, north Africa, the Middle East, central Asia, and west Africa | 230,000–430,000 CL |
| Leishmania (Leishmania) infantum | VL and CL | Persistent fever and splenomegaly in VL; typically single nodules and minimal inflammation in CL | VL is fatal within 2 years if untreated; CL lesions self-heal within 1 year conferring individual immunity | Children under 5 years and immunocompromised adults for VL; older children and young adults for CL | Dogs, rodents, rabbits and hares, foxes, opossums, and humans | Peridomestic zoonotic | Phlebotomus Larroussius subgenus (e.g., P. ariasi, P. Perniciosus), Lutzomyia longipalpis, Lutzomyia olmeca | OW and NW | China, southern Europe, Brazil, and South America for VL and CL; Central America for CL | 6200–12,000 cases of Old World VL and 4500–6800 cases of New World VL; unknown number of CL cases |
| Leishmania (Leishmania) mexicana | CL, DCL, and DsCL | Ulcerating lesions, single or multiple | Often self-healing within 3–4 months | No well-defined risk groups | Rodents and marsupials | Sylvatic zoonotic | Lutzomyia olmeca, Lutzomyia ayacuchensis | NW | South America | Limited number of cases, included in the 187,200–300,000 total cases of New World CL |
| Leishmania (Leishmania) venezuelensis | CL | Ulcerating lesions | Not well described | No well-defined risk groups | Unknown | Zoonotic | Lutzomyia olmeca? Lutzomyia bicolor? | NW | Venezuela | Limited number of cases, included in the 187,200–300,000 total cases of New World CL |
| Leishmania (Leishmania) amazonensis | CL, DCL, and DsCL | Ulcerating lesions, single or multiple | Not well described | No well-defined risk groups | Opossums and rodents | Sylvatic zoonotic | Lutzomyia flaviscutellata | NW | South America | Limited number of cases, included in the 187,200–300,000 total cases of New World CL |
| Leishmania (Viannia) braziliensis | CL, MCL, DCL, and LR | Ulcerating lesions can progress to mucocutaneous form; local lymph nodes are palpable before and early on in the onset of the lesions | Might self-heal within 6 months; 25% of cases progress to MCL | No well-defined risk groups | Dogs, humans, rodents, and horses | Sylvatic zoonotic | Lutzomyia wellcomei, Lutzomyia migonei, Lutzomyia neivai, Lutzomyia carrerai | NW | South America | Limited number of cases, included in the 187,200–300,000 total cases of New World CL |
| Leishmania (Viannia) guyanensis | CL, DsCL, and MCL | Ulcerating lesions, single or multiple that can progress to mucocutaneous form; palpable lymph nodes | Might self-heal within 6 months | No well-defined risk groups | Opossums, sloths, and anteaters | Sylvatic zoonotic | Lutzomyia whitmani, Lutzomyia shawi, Lutzomyia anduzei, Lutzomyia ayacuchensis | NW | South America | Limited number of cases, included in the 187,200–300,000 total cases of New World CL |
| Leishmania (Viannia) peruviana | CL | Ulcerating lesions, single or multiple | Not well described | No well-defined risk groups | Unknown, dogs? | Zoonotic | Lutzomyia peruensis, Lutzomyia verrucarum | NW | Peru, Bolivia | Limited number of cases, included in the 187,200–300,000 total cases of New World CL |
| Leishmania (Viannia) panamensis | CL, MCL and DCL | Ulcerating lesions, single or multiple that can progress to mucocutaneous form | Not well described | No well-defined risk groups | Rodents, dogs? | Sylvatic zoonotic | Lutzomyia gomezi, Lutzomyia hartmanni, Lutzomyia trapidoi, Lutzomyia yuilli | NW | Central and South America | Limited number of cases, included in the 187,200–300,000 total cases of New World CL |
| Leishmania (Viannia) lainsoni | CL | Ulcerating lesions, single or multiple | Not well described | No well-defined risk groups | Rodents, porcupines | Sylvatic zoonotic | Lutzomyia ubiquitalis | NW | Brazil, Bolivia, Peru, Ecuador | Limited number of cases, included in the 187,200–300,000 total cases of New World CL |
| Leishmania (Viannia) lindenbergi | CL | Ulcerating lesions | Not well described | No well-defined risk groups | Unknown | Zoonotic | Lutzomyia atunesi? | NW | Brazil | Limited number of cases, included in the 187,200–300,000 total cases of New World CL |
| Leishmania (Viannia) naiffi | CL | Ulcerating lesions, single and small | Not well described | No well-defined risk groups | Rodents, Anteaters | Sylvatic zoonotic | Lutzomyia ayrozai, Lutzomyia squamiventris | NW | Brazil, French Guyana | Limited number of cases, included in the 187,200–300,000 total cases of New World CL |
| Leishmania (Viannia) shawi | CL | Ulcerating lesions | Not well described | No well-defined risk groups | Rodents, sloths | Sylvatic zoonotic | Lutzomyia whitmani | NW | Brazil | Limited number of cases, included in the 187,200–300,000 total cases of New World CL |
| Leishmania colombiensis# | CL and VL | Not well described | Not well described | No well-defined risk groups | Sloths | Sylvatic zoonotic | Lutzomyia hartmanni | NW | Colombia | Limited number of cases |
| Leishmania (Mundinia) martiniquensis | CL and VL | Not well described | Not well described | No well-defined risk groups | Unknown | Likely sylvatic zoonotic | Unknown | OW and NW | Martinique, Thailand, Central Europe, USA | Limited number of cases |
| Leishmania (Mundinia) orientalis | CL and VL | Not well described | Not well described | No well-defined risk groups | Unknown | Likely sylvatic zoonotic | Unknown | OW and NW | Thailand | Limited number of cases |

Adapted from[7–9,31,123–128]. Legend: CL cutaneous leishmaniasis; DCL diffuse cutaneous leishmaniasis; DsCL disseminated cutaneous leishmaniasis; LR leishmaniasis recidivans; MCL mucocutaneous leishmaniasis; NW New World; OW Old World; PKDL post-kala-azar dermal leishmaniasis; VL visceral leishmaniasis. Notes: &For a more comprehensive list of vectors, including suspected ones, please check two previous Review Articles[7,31]. # L. colombiensis has been included in the genus Endotrypanum[129].

are easier to explain[14–17], the same is not true considering disease in immunocompetent individuals, suggesting that the parasite, vector, and host determinants that condition infection and/or disease are largely unknown. This said, each *Leishmania* species is associated primarily with one type of disease, not many incriminated vectors, and frequently qualified as either dermotropic or viscerotropic (Table 1). This aligns with the notion that leishmaniasis endemicity depends on active and sustained parasite transmission. Since neither parasite nor vector species are ubiquitous, it is not surprising that specific parasite species are associated with specific areas of the globe and, consequently with particular vector species of the same defined areas, that are permissive to infection (Table 1)[7,8].

Adding up to these notions, most of the *Leishmania* species that are pathogenic to humans are associated with zoonotic transmission. Only for *Leishmania donovani* parasites, no animals other than man have been incriminated as a reservoir [although evidence suggests that domestic dogs and mongooses (*Herpestes ichneumon*) may be reservoirs of this parasite species, as per studies from India[18] and East Africa[19,20]; for all of the remaining *Leishmania* species that cause disease in humans at least one animal reservoir (frequently sylvatic) is recognized (Table 1)[8,21]. All of the abovementioned justify the epidemiological complexity of leishmaniasis, whose control is, consequently, extremely difficult to accomplish[22]. In fact, there is a real risk of the disease(s) spreading to non-endemic regions, a consequence of the arrival of parasites and/or vector species to new areas, driven by changes in weather patterns and/or migrations[21,23,24]. It is, therefore, not unreasonable to qualify global warming, globalization, and war/conflicts as major potential risk factors of leishmaniasis emergence[21,25–27].

Therefore, the complete understanding of leishmaniasis (as a "whole"), depends not only on the dissection of the clinical aspects (parasite-host interactions) but also on the comprehension of the sand fly vectors and their interactions with *Leishmania* parasites and the animal/human hosts. However, most laboratories around the globe focus exclusively on parasite-host interactions, disregarding the sand fly vectors. In fact, information in the literature on sand flies is not easily accessible, at least in a comprehensible fashion. Therefore, the familiarization of new researchers with the vectors of *Leishmania* parasites can be a challenging task. To address this issue, here, we compile the basic information on sand flies, including the taxonomy (at a glance), biology, distribution, and life cycle, the blood-feeding process, and the *Leishmania*-sand fly interactions important for parasite transmission, as a resource for the scientific community in general. Moreover, we also discuss the outstanding questions in the field, answers to which are essential for the complete understanding of the parasite-vector-host interactions that lead to leishmaniasis

## Taxonomy at a glance, biology, distribution, and life cycle

Sand flies are arthropods and insects included in the order Diptera (two-winged flies), suborder Nematocera, family Psychodidae, and subfamily Phlebotominae[7]. Around 1000 sand fly species/subspecies were validated/described thus far around the world[28]. Initially, the taxonomical classification of sand flies was based on morphological analyses, including first an external analysis also known as phlebotometry (e.g., observation of the male genitalia, and determination of the wing venation indices…), and then the investigation of internal structures such as the spermathecae, cibarium, and the pharynx[7,29]. More recently, modern methods including chromosome analysis, isoenzyme analysis, molecular and phylogenetic analyses (DNA barcoding and Next-Generation Sequencing), and mass spectrometry

(MALDI-TOF), allowed the better identification and classification of sand fly specimens and consequently the clarification of some variations within sand fly subgenera/populations[7,30]. Many classification systems have been proposed over the years including those of Abonnenc, Davidson, Fairchild, Galati, Leng, Lewis, Quate, Rispail & Légerand, Secombe, Theodor, and Young & Duncan[7,28,30,31]. However, with respect to taxonomy and the classification of sand flies, there is still no consensus, reason why we decided not to describe each of the aforementioned classification systems in this Review; for more details, as well as a historical perspective on the taxonomy and systematics of sand flies, please check a few comprehensive reviews on the subject[7,28,30,31]. Instead, for the sake of simplicity, here we adopted the subdivision of the Phlebotominae into six genera, as per the widely accepted classification based on a conservative approach: *Phlebotomus* (13 subgenera), *Sergentomyia* (10 subgenera), and *Chinius* (four species) from the Old World, and *Lutzomyia* (26 subgenera and groups), *Brumptomyia* (24 species), and *Warileya* (six species) from the New World (Fig. 1)[7,30]. Of note, female sand flies (with the exception of a few autogenous species), apart from plant sugars (for flight energy and longevity), need to take a blood meal in order to develop and lay eggs. Importantly, the genera *Lutzomyia* and *Phlebotomus* are the ones that include the anthropophagous species (although some *Sergentomyia* may also feed on humans), and therefore, those relevant for the transmission of human disease (Table 1)[29].

In contrast to mosquitoes and other Diptera, sand flies do not have an aquatic stage in their life cycle[32]. Still, humidity is an important factor that together with temperature, are detrimental for, and influence sand fly development[33–35]. This justifies the fact that the sand fly distribution is limited to areas having temperatures above 15.6 °C for at least three months of the year[32], which still corresponds to the greatest portion of the world—from latitude 50° N to latitude 40° S (although they are absent from New Zealand and the Pacific islands) (Fig. 1)[36].

The sand fly life cycle comprises four major stages: eggs, larvae, pupae, and adults (Fig. 2). On average, a female sand fly deposits 30 to 70 eggs in protected places chosen based on humidity and the presence of organic matter (e.g., cracks and holes in the ground, animal burrows/dens, termite mounds, leaf litter…)[37,38]. Typically, the eggs hatch between four and 20 days after oviposition, although this timing may be extended in cooler weather—eggs may diapause under unfavorable conditions[38,39]. There are four larval instars, and larval development is usually completed in 20–30 days, depending on the sand fly species[40], as well as on the temperature and availability of food. However, this period may be prolonged to several months in sand fly species that diapause to cope with winter (only full-grown larvae diapause—instar four)[37,38]. Pupation usually takes from six to 13 days with adults emerging during the hours of darkness, often just before dawn[38]. Males usually emerge before females. Adult life expectancy in the wild has hardly been determined for sand flies, particularly considering females. However, it is known that while males may live only about a week in the wild, females may live longer as they undergo more than one gonotrophic cycle (some as many as three)[41]. Normally, oviposition occurs between five and eight days after blood-feeding, although some species are known to feed multiple times before successfully developing viable eggs[41]. Of note, most sand fly species are exophagic (feed outside of dwellings), although some are known to be endophagic and endophilic (feeding and resting in human and animal dwellings), commonly referred to as domestic or peridomestic species. A more detailed description of sand fly biology, behavior, and morphology (used to distinguish sand fly species) can be found elsewhere[29–31,37,38,41]. Still, much remains to be learned regarding these subjects.

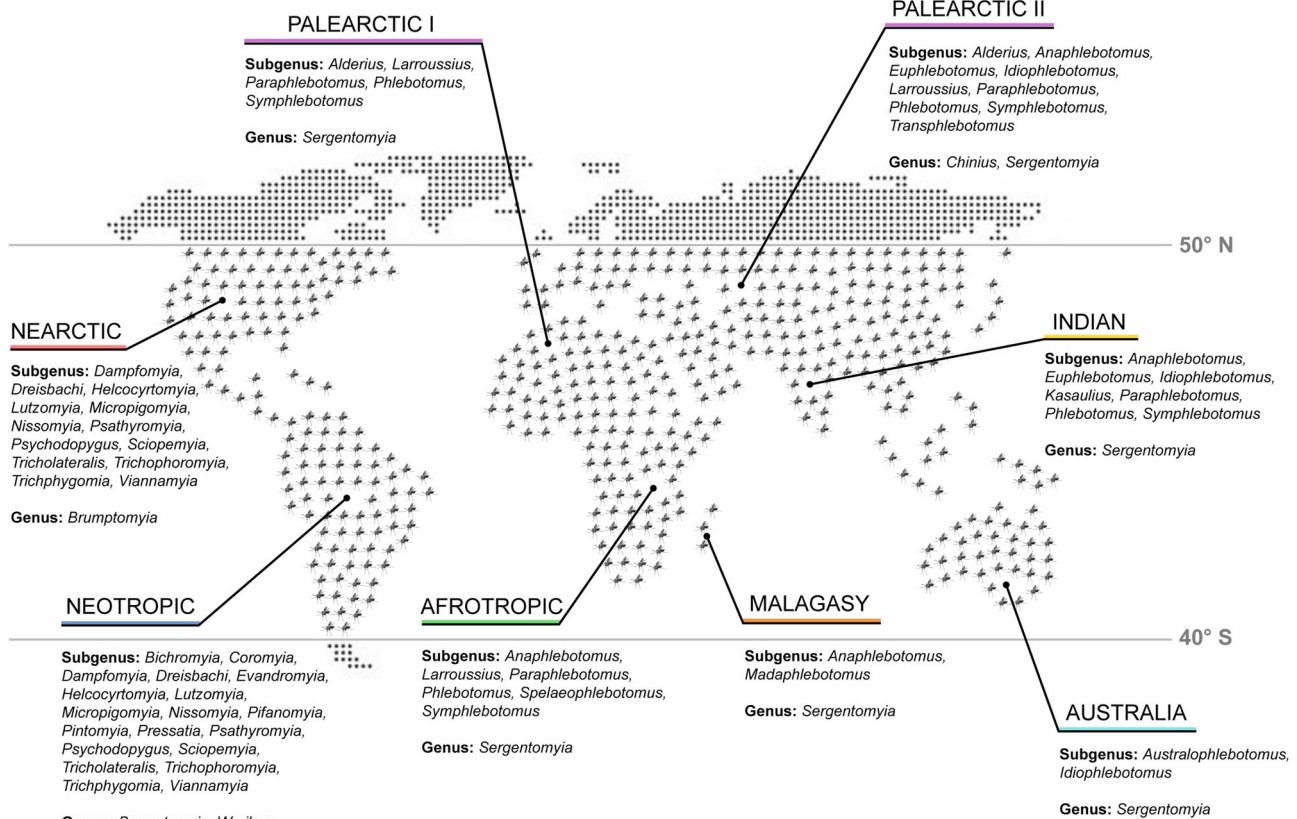

**Fig. 1 Sand fly distribution map by genera/subspecies.** Sand flies have a global distribution between latitude 50° N and latitude 40° S (demarked by the gray horizontal lines), excluding New Zealand and the Pacific islands. In the map, the relevant sand fly genera/subspecies (as per the widely accepted classification based on a conservative approach) are listed based on their presence in defined zoogeographical regions: Palearctic (purple), Nearctic (red), Neotropic (dark blue), Afrotropic (green), Malagasy (orange), Australia (light blue), and Indian (yellow). Adapted from[7]. Courtesy NIAID.

## The quest for blood and the blood-feeding process at a glance

The need for blood to give rise to a new sand fly generation (for the perpetuation of the species), associated with the fact that sand flies are weak fliers (reports state that adults usually disperse 100 meters or less from their larval habitats[32]), makes feeding-preferences relevant only in the context of high (and diverse) host abundance[38]. In other words, most times, feeding depends on host availability; sand flies will take blood from the closest permissive available source (although the engorgement outcome may vary)[32,42]. Of note, this aligns with the many times repeated idea that humans are generally accidental *Leishmania* hosts (obviously excluding those infected with the anthroponotic *L. donovani* parasites)[43,44]. Interestingly, contrarily to mosquitoes which are usually vessel feeders, sand flies are blood-pool feeders. They use their toothed mandibles in a scissors-like manner to lacerate the host's skin, disrupting cells and causing an extravascular pool of blood from which they ingest the blood meal[29,45]. Importantly, during this process, through salivation, sand flies introduce several pharmacologically active molecules into the skin, to facilitate feeding. The sand fly salivary proteome, usually less than 40 secreted proteins according to transcriptomic studies, is quite diverse in function[46]. While a vasodilator molecule promotes the increase of local blood circulation, a molecule with apyrase function inhibits platelet aggregation through the destruction of the agonist adenosine diphosphate (ADP), and together with molecules that inhibit the blood coagulation cascade and the classical pathway of the complement system, counteract an efficient hemostatic response[46]. Additionally, other proteins have relevant immunomodulatory properties, that are, however, not important for the feeding process[46].

## Sand flies as vectors of multiple diseases

Although sand flies are mostly recognized as *Leishmania* vectors, they transmit other pathogens, such as bacteria and viruses. Carrion's disease is a sand fly transmitted biphasic illness caused by *Bartonella bacilliformis* bacteria in Central/South America[47,48]. It is characterized by either intermittent febrile states (called Oroya fever), sometimes with hepatic involvement that can lead to death in the absence of (or delayed) treatment (when infected individuals are naïve); or by cutaneous lesions called Peruvian warts (when infected individuals were previously exposed to the bacteria)[48]. Some arboviruses are also human pathogens transmitted by sand flies, particularly those belonging to the *Phlebovirus* genus (order Bunyavirales, family Phenuiviridae)—enveloped single-stranded RNA(-) viruses[49]. At least nine virus "species" are recognized, containing 70 antigenically distinct viruses; of note, 33 other viruses, yet to be classified, are not included in the previous numbers[50]. Most of the sand fly transmitted viruses cause uncomplicated to moderate fever states known (when identified) as "sand fly fever" or "pappataci fever". However, one particular agent, Toscana virus has a marked tropism for the central and peripheral nervous systems, potentially causing neuro-invasive conditions such as meningitis or encephalitis[49,51–53].

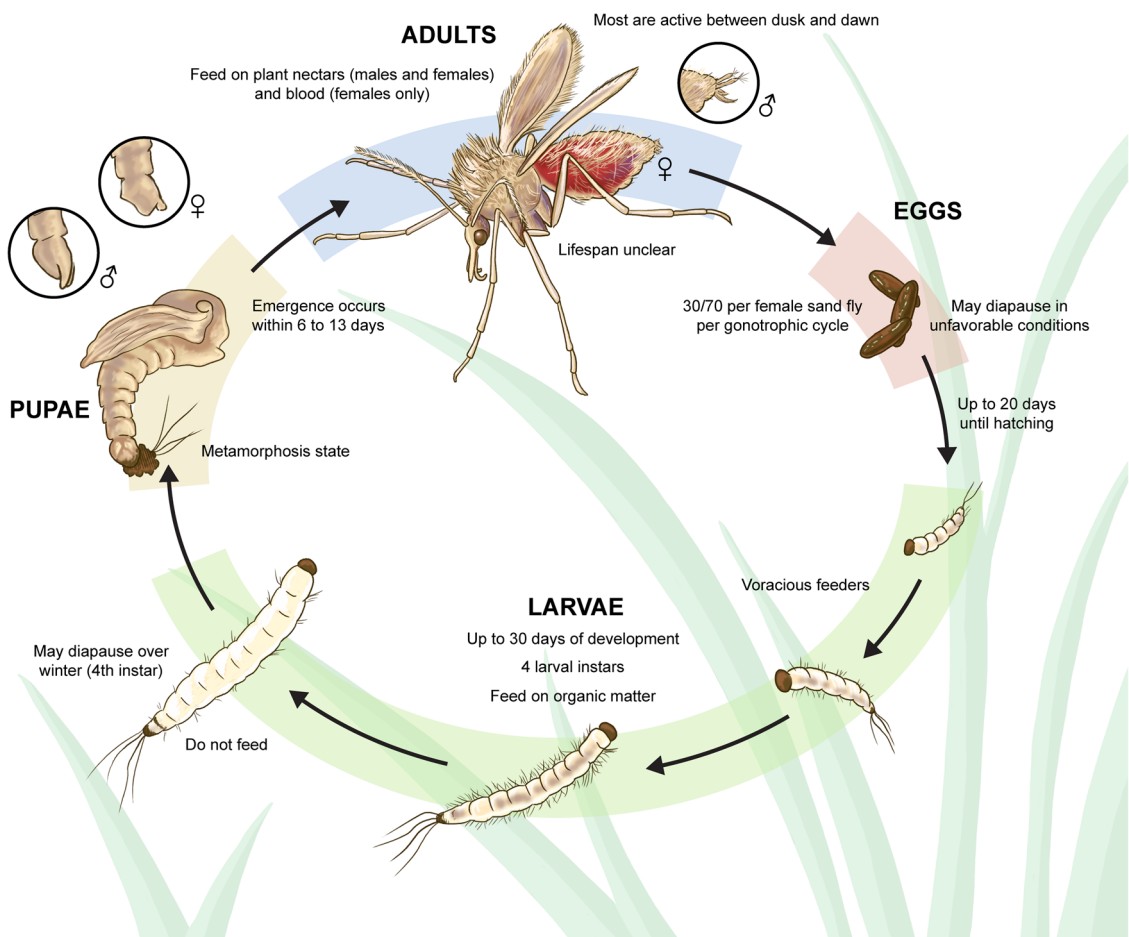

**Fig. 2 Schematic representation of the sand flies' life cycle.** The sand fly life cycle comprises four major stages: eggs (orange background), larvae (four instars: green background), pupae (yellow background), and adults (blue background). In the latter two stages, different morphological features (highlighted within the circles) can be used to distinguish the gender. The most important characteristics with respect to each stage (sub-stage), are listed near the images, as are the average timings of development. Adapted from[7,130,131]. Courtesy NIAID.

### Sand flies as vectors of leishmaniasis: permissive *versus* restrictive

Among the 1000 sand fly species/subspecies validated/described thus far around the world[28], only one-tenth (10%) are proven or suspected vectors of *Leishmania* parasites[31,54]. These meet all (proven vectors) or almost all of the vector incrimination criteria proposed by Killick-Kendrick[55] and the WHO Expert Committee on the control of Leishmaniases[56]: (i) they feed on humans (are anthropophilic), (ii) they also feed on the relevant reservoir hosts in the case of zoonotic agents, (iii) they are found in nature infected with the same parasites (*Leishmania* species) circulating in humans (from the same geographical area); (iv) they support the complete development of the *Leishmania* parasites circulating in humans, including after the defecation of the bloodmeal remnants; and (v) they are able to transmit those parasites to susceptible hosts when they take a bloodmeal[30,31]. Importantly, with respect to the sand fly vectors incriminated thus far, *Leishmania*-sand fly interactions studied under laboratory conditions led to their separation into two major groups: restrictive and permissive vectors[57]. As the names indicate, while the first group displays a remarkable specificity for the *Leishmania* species they transmit in nature (e.g., *Phlebotomus papatasi* and *Phlebotomus sergenti*), the second permits the development of distinct *Leishmania* species (e.g., *Phlebotomus arabicus* and *Lutzomyia longipalpis*)[57]. For instance, recently, we reported that *L. longipalpis* sand flies, vectors of *Leishmania infantum* parasites in

nature, are competent vectors of *Leishmania major* parasites under laboratory conditions. We demonstrated that *L. longipalpis* sand flies are able to acquire *L. major* parasites from cutaneous leishmaniasis active lesions, to sustain mature infections, and to transmit the parasites to naïve hosts, causing disease[58]. This permissive *versus* restrictive dichotomy is thought to be related to parasite attachment to the sand fly midgut, defined as an essential mechanism for infections to proceed within the sand fly—discussed in detail in the following section[59,60]. However, while we understand these interactions in restrictive vectors (mediated by very specific ligand-receptor interactions), a lot is yet unknown regarding permissive ones[59]. What we know in this regard considering restrictive (or specific) vectors comes from the study of *L. major* development within *P. papatasi* sand flies. In this context, the attachment of parasites to the midgut is mediated by the binding of *L. major* lipophosphoglycan (LPG) molecules to a specific sand fly midgut receptor, a galectin (β-galactoside binding family of lectins)[61]. Importantly, although LPG molecules are very abundant surface proteins, found in all *Leishmania* species, they are also polymorphic (particularly the 10–30 phosphoglycan repeating units) and different, not only considering parasite species but also parasite strains, and even stages[61,62]. Interestingly, these differences explain both vector restrictiveness and the attachment-detachment processes of *L. major* parasites to the midgut of *P. papatasi*, obviously dependent on the specificity of ligand–receptor interactions[61]. On the other hand, vector

permissiveness suggests broader or even non-specific binding processes. Although these processes are yet to be fully understood, the involvement of sand fly midgut "sticky proteins" is a considered hypothesis (e.g., O-linked glycoproteins with mucine-like properties)[63].

### Sand fly-*Leishmania* interactions toward the development of mature infections

The development of *Leishmania* parasites within the sand fly vector is quite complex (Fig. 3), with distinct differentiation processes that are required for the establishment of a successful infection. Of note, contrary to many vector-borne agents (including some Trypanosomatids), the development of *Leishmania* parasites is confined to the sand fly digestive tract (there is no crossing/disruption of the epithelial barrier[59,64]), simplistically divided here (excluding the crop) into: (i) the foregut— the most anterior portion, from the mouth to the cardia, which includes the stomodeal valve; (ii) the (thoracic and abdominal) midgut— from the cardia to the pylorus; and (iii) the hindgut—the most posterior portion, from the pylorus to the rectum[65].

Although most *Leishmania* species are suprapylarian (development restricted to the midgut), species of the subgenus *Viannia* colonize the hindgut before migrating forward to the midgut (peripylarian parasites)[65]. The first differentiation step occurs not long after ingestion of the infected blood meal by sand flies. Due to changes in conditions (such as the decrease of temperature and the increase in the pH), the amastigotes differentiate into procyclic promastigotes, weakly motile forms[59,66]. This first differentiation step was proposed to be extremely important; it was postulated that parasites within the blood meal need to resist the effect of digestive proteases, the first and one of the most significant barriers to parasite survival[67]. In one study, *L. major* procyclic promastigotes were demonstrated to be more resistant to proteolytic attack than the parasites in the transitional state (from amastigote to promastigote forms)[68]; a possible explanation for such a phenotype is the known dynamic changes of the parasites' glycocalyx components: e.g., comparing amastigotes with promastigotes, the latter have a higher content of LPG in their membrane[62,67]. On the other hand, a more recent study showed contrary findings, and the authors suggested parasite killing (*L. major* and *L. donovani*) could be due to the toxic products of blood meal digestion[69]. Importantly, although contradictory, these studies both suggest the impact of midgut proteases in the establishment of *Leishmania* parasites within the vector (either directly or indirectly); this notion is further supported by studies focusing on other vector-parasite pairings (*Lu. longipalpis* – *L. mexicana*/*L. infantum*) showing that *Leishmania* parasites thrive after the downregulation of the proteolytic activity in the sand fly midgut[70–72]. Of note, procyclic promastigotes are also the first replicative form found within the sand fly vector, increasing parasite numbers, important to the next step of the vector infection cycle[59].

Within the sand fly midgut, the blood meal is enveloped by a type I peritrophic matrix. This structure, found in blood-fed insects, has mainly a semi-barrier protective role: (i) against the possible damage caused to midgut microvilli by the concentrated digestive environment; (ii) against the potentially devastating effects of one of the blood digestion by-products, heme; and (iii) against potential pathogens[73]. *Leishmania* parasites must "escape" from this structure in order to establish an infection. Around 48 h after the blood meal ingestion, procyclic promastigotes begin to slow their replication and differentiate into strongly motile and long forms called nectomonad promastigotes. These developmental forms are the ones that "escape" from the peritrophic matrix into the midgut lumen[59]. Furthermore, these

parasite forms are also responsible for the attachment to the sand fly midgut, another crucial step for the completion of the *Leishmania* cycle within the vector, since it prevents the parasites from being eliminated together with the blood meal remnants during defecation[59,74,75]. Of note, even in refractory vectors parasites can differentiate into promastigotes and multiply within the blood bolus, but are then eliminated via defecation[69,74]; therefore, the "escape" from the peritrophic matrix and the attachment to the sand fly midgut to avoid the defecation-mediated elimination are detrimental for the establishment of *Leishmania* parasites within permissive sand flies. In line with this, it is important to repeat here one of the criteria that need to be met for the incrimination of sand flies as vectors of *Leishmania* parasites— sand flies support the complete development of the *Leishmania* parasites circulating in humans, including after the defecation of the bloodmeal remnants—and stress the fact that the detection of parasites or their DNA in engorged sand fly females (before the defecation) is insufficient for vector incrimination[76].

The life cycle then continues with the migration of nectomonad promastigotes towards the anterior midgut and their differentiation into leptomonad promastigotes. These shorter parasite forms are another replicative stage in the insect, responsible for the population of the sand fly anterior midgut as well as for the secretion of the promastigote secretory gel (PSG), important for the transmission process[77,78]. Eventually, the leptomonads undergo another differentiation step called metacyclogenesis giving rise to the parasite stage infective to vertebrate hosts, the metacyclic promastigote[59,64]. Although the metacyclogenesis determinants are yet to be fully understood, the nutritional deprivation hypothesis (and probably a resultant quorum sensing mechanism) makes sense and is supported by at least one study that shows that the absence of purines promotes parasite differentiation into metacyclic forms[79]. Metacyclic parasites have a smaller body, and a long flagellum, responsible for their extremely fast motility[64]. Additionally, leptomonad promastigotes are also thought to attach to the sand fly stomodeal valve and give rise to haptomonad promastigotes, the less studied (and thus the most "neglected") vector-derived parasite form, whose role is not completely clear[80]. This said, the attachment of these parasite forms may be important for the loss of the integrity of the stomodeal valve (together with the action of parasite-derived chitinolytic enzymes), which is relevant for the transmission process[81–83]. Importantly, more than morphologically, all of these parasite stages were demonstrated to be transcriptionally distinct to varying degrees[84].

In nature, sand flies are expected to take a blood meal every five to six days, to complete as many gonotrophic cycles as possible[85]. Consequently, the nutrient-deprived environment that develops in the sand fly midgut as the *Leishmania* infection progresses, is expected to be transient. This notion consequently changes the traditionally envisioned *Leishmania* cycle within the vector (linear), which needs to be adapted to the ecological reality (with an associated dynamicity). Serafim et al. showed the consequences of a second blood meal (non-infected) in experimental sand fly infections. The most important one led to the breaking of the dogma that metacyclics are the last stage in the life cycle of *Leishmania* within the vector. The revisited life cycle within the vector includes a new parasite stage called retroleptomonad, originated by the de-differentiation of metacyclic parasites in the presence of newly available nutrients, a consequence of blood intake by infected sand flies[86]. As the name implies, this newly described parasite form is morphologically closer to leptomonad promastigotes, as it appears to be functionally: contrary to metacyclics and similar to leptomonad promastigotes, retroleptomonads are replicative forms[86]. Eventually, when the stress conditions are re-established in the midgut (with the defecation of

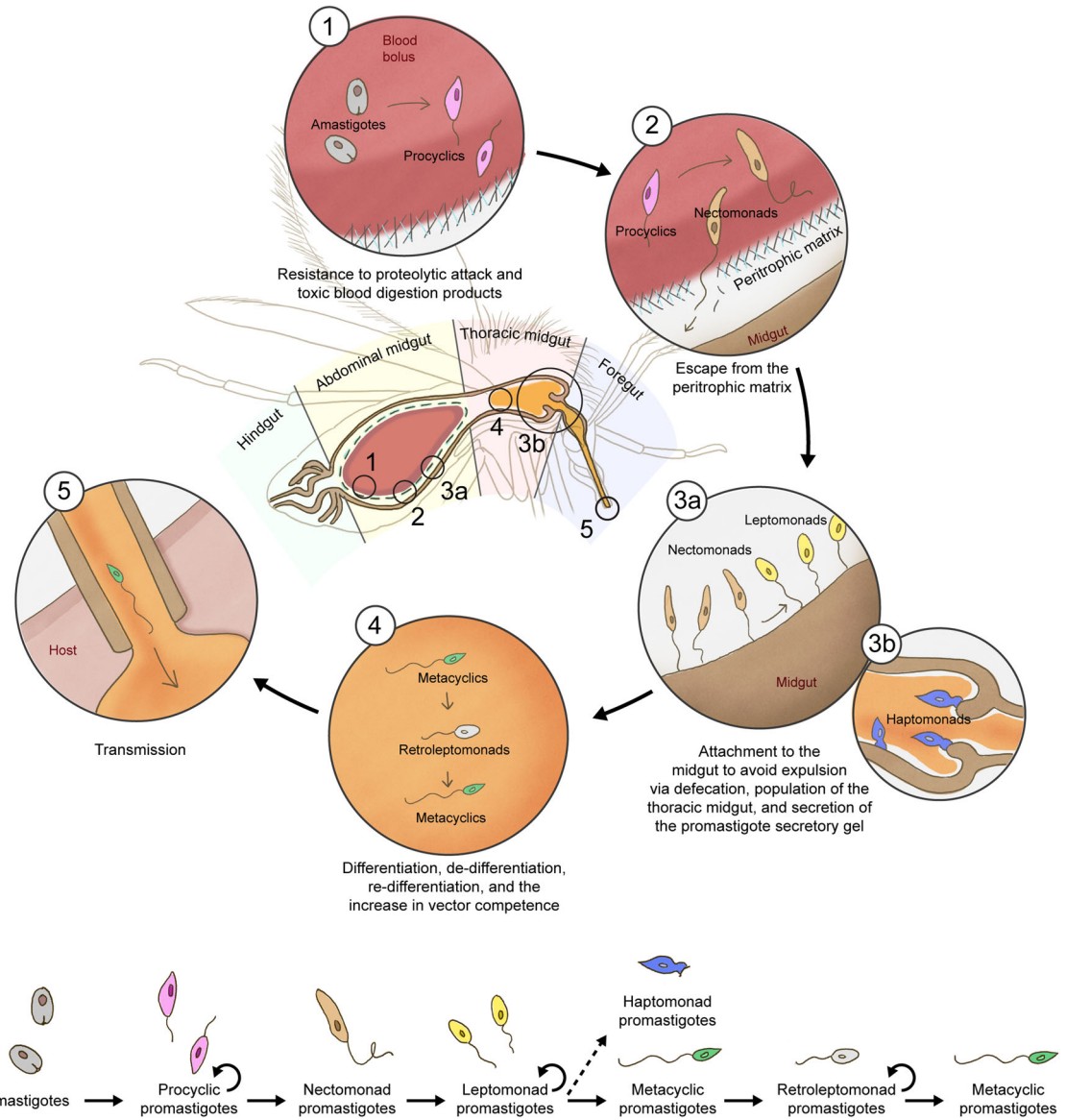

**Fig. 3 *Leishmania* development within the sand fly midgut.** Schematic representation of the different forms of *Leishmania* parasites within the sand fly vector and of the major barriers they must overcome to establish a productive infection—including the resistance to proteolytic attack/toxic byproducts of the digestion of blood (1), "escape" from the perothrophic matrix (2), attachment to the midgut to avoid expulsion (3a), attachment to (and impairment of) the stomodeal valve (3b), and (de-)differentiation and replication dynamics (4)—and ensure their transmission (5) to a new host. A linear life cycle with the different parasite forms within the vector is also represented; the circular arrows highlight the replicative parasite forms. Adapted from[60,86]. Courtesy NIAID.

the second blood meal remnants), retroleptomonad parasites "re-differentiate" into metacyclic promastigotes[86]. Importantly, the retroleptomonad replication has as a consequence, better vector infections, both quantitatively (increased parasite numbers per midgut) and qualitatively (more homogeneous populations of metacyclic promastigotes) (Fig. 4)[86].

The above-mentioned findings, aligned with the ecological context, may suggest that the development of a successful infection in wild sand flies is a gradual process, dependent on parasite amplification boosted by the intake of multiple blood meals by the sand fly vector (Fig. 4)[86]. Even if initially the parasite numbers are very few, the intake of a second blood meal will boost the replication of leptomonads, increasing parasite numbers up to a point that favors their differentiation into metacyclics; a third blood meal and another round of parasite replication may be necessary, depending on the initial infectious inoculum. Still,

this may not be enough to originate a productive infection, known to be dependent on the infectious inoculum (at least experimentally)[86,87]. Nevertheless, the de-differentiation of metacyclic promastigotes into replicative retroleptomonads upon the intake of subsequent blood meals by infected sand flies will potentiate the development of "better" infections and increase vector competence up to a point that transmission is the most likely scenario (Fig. 4), assuming the sand fly survives long enough.

## *Leishmania* transmission: the infectious inoculum

The deposition of metacyclic parasites into the host's skin is dependent on their regurgitation by the sand fly vector. Importantly, this process is probably the result of a clever adaptation of *Leishmania* spp. parasites. The filamentous proteophosphoglycans

**Fig. 4 The impact of multiple blood-meals on the maturation of *Leishmania* infections within the sand fly vector.** In nature, sand flies are expected to feed on blood multiple times for the completion of more than one gonotrophic cycle. Importantly, the intake of multiple bloodmeals (represented by the red blood drops) is expected to impact the vector competence, promoting not only the increase in the absolute parasite numbers (yellow) but also in both the percentage and number of the metacyclic infectious forms (green) in the sand fly midgut. Importantly, such an increase in total parasite numbers (**A**), and particularly in the number of metacyclic promastigotes (**B**) in the midgut of infected sand flies that take subsequent blood meals, compared with single-fed flies (blue lines), results in a higher probability of transmission of *Leishmania* parasites (purple gradient). Courtesy NIAID.

secreted by leptomonad promastigotes form a gel-like plug called PSG plug, as mentioned above, that impairs the sand fly feeding process. Because the infected sand fly digestive tract is clogged, to an extent dependent on the parasite burden (the more parasites, the bigger the plug[86]), to facilitate blood intake (trying to unclog the anterior midgut), sand flies regurgitate[8]. Importantly, such a regurgitation is also facilitated by the *Leishmania*-induced damage of the sand fly stomodeal valve, known to be permanently opened in the context of heavy infections[81,83]. Of note, as an indirect consequence of the formation of the PSG plug/damage of the stomodeal valve, the behavior of infected sand flies is also altered toward an increase in the feeding persistence, with the potential to favor infection. Researchers have shown that heavily infected flies (with larger PSG plugs - "blocked sand fly" phenotype) had more difficulty in taking a full blood meal, and thus attempted to re-feed more often and on multiple hosts, positively impacting transmission[78,88]. Having the above in mind, in the end, mostly metacyclic promastigotes are egested into the skin of (multiple) hosts (in the context of a mature sand fly infection), but not alone. We now know that the infectious inoculum is composed of many relevant factors, both parasite- and vector-derived.

The *Leishmania*-derived proteophosphoglycans, part of the PSG plug, that is regurgitated together with parasites, were demonstrated to contribute to disease exacerbation (in the context of both cutaneous and visceral leishmaniasis—CL and VL, respectively)[67]. A possible mechanism proposed was the modulation of early innate pathways involved in response to a wound. The PSG was shown to potentially accelerate wound healing in the skin[89], which in turn is known to be by itself a potential infection-enhancer stimulus[90,91]. Additionally, parasite-derived exosomes were also shown to be part of the infectious inoculum, and to potentiate disease, in both CL and VL animal models[92,93]. The alteration of cell recruitment patterns, and the modulation of cell behavior, were mechanisms shown to be involved in the exosome-mediated potentiation of infection[92,93].

With respect to the vector-derived infection enhancers, both the sand fly gut microbiota and sand fly saliva were demonstrated to play a role[94–96]. Similar to almost every known gastrointestinal tract in nature, sand fly midguts harbor a diverse microbiological community. Importantly, the colonization of the vector intestinal tract was shown to be extremely important for the development of *Leishmania* parasites within the sand fly midgut[97]. Additionally, these sand fly bacterial midgut colonizers were shown to be co-egested together with *Leishmania* parasites and the above-mentioned parasite-derived infection enhancers during transmission, additively contributing to infection establishment[98]. The egested microbes were shown to trigger the inflammasome, leading to a rapid production of IL-1β and a sustained neutrophil infiltration at the site of the vector bite[98]. Of note, the essential role of neutrophils in the context of *Leishmania* infection-

establishment was shown for *L. major* parasites more than a decade ago[99], although some studies point to an infection-protective role[100]. Last but not least, all of these midgut-derived immunomodulatory factors (and *Leishmania* parasites) will join the sand fly saliva within the host skin. And more than to impact host hemostasis, sand fly saliva was shown to modulate host immunity. Sand fly salivary proteins were shown to favor (in most contexts) anti-inflammatory local immune responses, beneficial for the establishment of parasites, and consequently, to potentiate infection and disease[101]. Of note, such an infection-enhancing effect of sand fly saliva was demonstrated in vivo both in the context of transmission (the establishment of infection)[102,103], and of active disease (cutaneous leishmaniasis mouse model)[104]. Interestingly, recently, a family of insect-derived neutrophil chemoattractant proteins was identified, for the first time, in the saliva of sand flies, with infection-enhancer characteristics[105].

## Outstanding questions

Over the past few decades, our understanding of sand fly vector–parasite–host interactions has considerably improved. However, many gaps in knowledge must still be addressed in the field, in leishmaniasis endemic areas.

Most of what we know about *Leishmania* transmission by sand flies is based on laboratory evidence. Therefore, the translation of these notions to the natural context, or, in other words, their validation is still needed. The clarification of the transmission determinants is essential. Many questions still need to be answered, such as "Is there a threshold of infection (both related to the prevalence of infected flies and the average infection burden) associated with effective transmission?", and "Can the bite of a single infected fly lead to the development of visceral leishmaniasis?". Of note, some interesting theories have been proposed[106]; however, they are again mainly supported by laboratory findings.

In fact, the ecological context is mostly unknown, and probably different considering the distinct vector–parasite–reservoir combinations found in nature. Only the dissection of such interactions considering each particular combination will enable us to fully understand disease transmission and either optimize the control strategies available or develop better-suited ones. With this respect, considering that most disease-causing *Leishmania* species are zoonotic agents, to investigate the interactions between sand flies and the (sylvatic) *Leishmania* reservoirs is of paramount importance. For instance, it is essential to understand whether in nature the interaction between sand flies and reservoirs boosts each other's competence, in a vicious cycle that ultimately leads to the perpetuation of *Leishmania* parasites, as was somehow suggested in a laboratory study[107].

Also related to the above topic the complete disclosure of the competent vectors at a given location (including the potential

incrimination of new sand fly species), is vital for the development of vector control strategies with real impact on disease control[108]. As a speculative exercise, thinking on the Mediterranean Basin, where at least eight different sand fly species were incriminated as vectors of *L. infantum* parasites[109], to focus the control strategy on the main vector species in the area, but not in all permissive sand fly species may have little/limited impact on disease control, considering the possibility of redundancy. Other vector species, either endemic or emergent [global warming is expected to change the sand fly distribution landscape[109,110]] in the region, may assume the role of "main disease vector".

Additionally, also on the topic of vector control, it is of paramount importance to disclose the immune responses in sand flies, in the context of *Leishmania* infection. And although more than a few breakthroughs were made on this topic in the last years, as recently reviewed in detail by Telleria and colleagues[96], much is yet unknown, particularly with respect to the existence of immune-related determinants of vector refractoriness. The fact that both the modulation of the gut microbiota[97,111,112] and infection with particular viral agents[113] impact the establishment of *Leishmania* parasites within the sand fly vector, makes us wonder whether more than the simple competition, this is due to some kind of "immune priming", as reported for other relevant vectors of human disease[114,115]; this hypothesis is worth to be explored in the future. Importantly, only when we comprehensively understand the sand fly immune responses detrimental for the elimination of *Leishmania* parasites, can we start trying to answer the question: can we modulate immunity in sand flies to make them refractory to *Leishmania* parasites and use this as a vector-based strategy for the control of leishmaniasis?

The deep knowledge of sand fly biology and behavior is also vital for the complete understanding of leishmaniasis and the rational development of prophylactic and even therapeutic interventions. Substantial progress has been made in this respect. For instance, the definition of the infectious inoculum discussed above is quite recent. Additionally, more and more sand fly-based vaccine candidates against leishmaniasis[116,117] are being proposed as essential disease control tools that can be used in combination with the *Leishmania*-derived ones[118,119]. The same is true considering the exploitation of sand fly salivary proteins as markers of exposure, as a tool for the control of leishmaniasis[46,95]. However, much still remains to be done; many questions are still unanswered. For instance, to what extent do vector-derived factors impact host immunity and how can we overcome such responses in a significant fashion? Additionally, should we revisit the aim of vector-derived vaccines - inducing Th1 DTH immune responses *versus* blocking the activity of infection-enhancer molecules including neutrophil chemoattractant proteins, hyaluronidases, and endonucleases[120,121]?

Last but not least, the acknowledgment of the vector as part of the equation is also indispensable for a complete understanding of leishmaniasis. As mentioned in the Introduction, most laboratories interested in leishmaniasis do not have access to sand flies. Therefore, most of the information generated may not be translatable to the natural context. A good example of the negative impact of the disregard of the vector in experimental studies was published by Peters et al.; these authors showed in vivo that a previously defined effective anti-*Leishmania* vaccine lost its protective potential in the context of natural *Leishmania* transmission via sand fly bites[122]. Therefore, one question that deserves to be answered is: "Are there any established dogmas based on the incomplete focus on the parasite–host interactions that are not valid in the context of the vector? Also in line with this limitation, we also dare to ask whether surrogate models (closer to the natural context than the traditionally used leishmaniasis in vitro/animal models) can be developed and widely employed?

## Conclusion

As a vector-borne disease, leishmaniasis is the result of an intricate web of vector–parasite–host interactions. However, while the hosts and the parasites are the traditional focus of the studies, the vectors are often overlooked. In this review, we tried to compile information, in our opinion, essential for new researchers to become familiarized with sand flies, in the context of leishmaniasis. Importantly, only when we understand sand flies as well as we do *Leishmania* parasites and their hosts, will we be able to establish the determinants of transmission and disease and to implement strategies to effectively control leishmaniasis.

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

## Acknowledgements

We are grateful to all of our laboratory colleagues for the stimulating discussions along the years. We are also grateful to Shaden Kamhawi (NIAID, NIH, USA), Claudio Meneses (NIAID, NIH, USA), Philip Lawyer (Brigham Young University, USA), Claudia Brodskyn (Fundação Oswaldo Cruz-CPqGM, Brazil), John Andersen (NIAID, NIH), and Janneth Rogrigues (GSK, Tres Cantos, Spain) for the valuable feedback. Finally, we acknowledge Rose Perry-Gottschalk (Research Technologies Branch, NIAID, NIH), who created the illustrations. This study received funding from the project NORTE-01-0145-FEDER-000012, supported by Norte Portugal Regional Operational Program (NORTE 2020), under the PORTUGAL 2020 Partnership Agreement, through the European Regional Development Fund - ERDF (A.C.-d.-S.). This study was also supported in part by the Intramural Research Program of the NIH, National Institute of Allergy and Infectious Diseases (F.O.). P.C. was supported by Foundation for Science and

Technology (FCT), Portugal, through the individual Grant SFRH/BD/121252/2016. P.C. was also supported by Fulbright Portugal. The funders had no role in study design, data collection and analysis, decision to publish, or preparation of the manuscript.

## Author contributions

P.C. and F.O. envisioned the structure of the manuscript. P.C. conducted the bibliographic search and wrote the initial draft of the manuscript. A.C.-d.-S. and F.O. reviewed and edited the manuscript. All authors listed have made a substantial, direct, and intellectual contribution to the work, and approved it for publication.

## Funding

Open Access Funding provided by the National Institutes of Health (NIH).

## Competing interests

The authors declare no competing interests.
