## [Peer Review File · Communications Biology]

Reviewers' comments:

Reviewer #1 (Remarks to the Author):

This manuscript is focused on Phlebotomine sand flies, vectors of medically important *Leishmania* parasites. The author's aim was to "familiarize new researchers with these insects" - they compiled information on *Leishmania* – sand fly interactions with information on sand fly biology, ecology and behaviour. While the chapters focused on *Leishmania* – sand fly interactions are written in sufficient details (with exception of some, further pointed items), the parts describing sand fly taxonomy, biology and ecology are reduced on very basic facts. It was, probably, the intention of authors but, in this case, I would expect references to recent papers or book chapters summarizing properly the biology of sand flies and their role in leishmaniasis transmission. To my surprise, references to two important recent reviews on sand fly biology, distribution and *Leishmania* transmission are missing (Maroli et al 2012, doi: 10.1111/j.1365-2915.2012.01034.x and Dvorak et al 2018, doi.org/10.1007/978-3-319-72386-0_1). Valuable is the last chapter summarizing the outstanding questions and pointing topics that should be answered to understand the complex interactions connected with leishmaniasis.

Specific comments:

1. According to the abstract, the authors should discuss the taxonomy of sand flies. However, this is not true; the taxonomy is provided on 5 rows only as a categorization of the subfamily Phlebotominae and a list of the 6 sand fly genera. This very simple introduction does not allow any deeper insight into a discussion on sand fly taxonomy. Authors should either change this sentence in the abstract or enrich the text on sand fly taxonomy (Does the current taxonomy reflect the phylogeny of the group? What are the main morphological traits important for sand fly species discrimination? Which molecular methods have been used and how improved species identification and sand fly taxonomy? ...)
2. Line 44: "more than 300 species feed on blood" the number is seriously underestimated; even bloodfeeding sand flies are almost 1000 species. Consider either changing it or deleting the entire sentence.
3. Line 76: The sentence „Evidence suggest that domestic dogs may be reservoirs of this parasite species "is supported with the citation of one manuscript (No.18) based on data from India. However, a role for animal reservoir hosts in the transmission of *L. donovani* has been suggested for many years in East Africa. Authors should mention this and cite at least two other sources: Elnaiem et al (2001) doi:10.1017/S0031182001007594, Dereure et al. (2003) doi.org/10.1016/j.micinf.2003.07.003s.
4. Lines 87-97: This part is very similar to Abstract (lines 31-38). Please, consider revision.
5. The chapter about taxonomy, distribution and life cycle could be renamed "Biology, distribution and life cycle". Two important reviews mentioned above should be referred.
6. Information about the life cycle and life tables on Lines 118-133: I am missing two valuable references on this topic, the first summarizing life table of about ten sand fly species under standard conditions (Volf and Volfova 2011, DOI 10.1111/j.1948-7134.2011.00106.x), and the second demonstrating the effect of temperature on sand fly life cycle (Benkova and Volf, 2007, DOI 10.1603/0022-2585).
7. The chapter dedicated to sand flies as vectors of *Leishmania* parasites should be introduced by the information that only 10% of described sand fly species are proven or suspected vectors of *Leishmania*. Authors should also provide criteria for vector competence specified by Killick-Kendrick (1990), doi.org/10.1111/j.1365-2915.1990.tb00255.x and WHO (2010) Control of the leishmaniasis. Report of a Meeting of the WHO Expert Committee on the Control of Leishmaniasis.
8. In the chapter "Sand fly – *Leishmania* interaction, authors should briefly describe the morphology of the sand fly digestive tract and define the foregut, midgut and hindgut and their functions before using these terms for an explanation of suprapylarian, peripylarian and hypopylarian development of parasites.
9. On lines 214-220 authors cite old hypotheses that proteases pose the first and one of most significant barriers to parasite survival and procyclic promastigotes are more resistant than transforming stages. However, more recent experiments did not show any direct negative effect of proteases on transforming parasites while documenting that leishmania mortality results from toxic products of blood meal digestion (Pruzinova et al 2018, DOI 10.1186/s13071-018-2613-2). Please, revise the text based on these findings.

10. In this chapter, authors should also emphasize the important fact that the early phase of *Leishmania* development in the vector is nonspecific. Before defecation, almost any *Leishmania* species survive also in sand fly species, which are not competent vectors (Pruzinova et al 2018, Sadlova et al 2018 doi.org/10.1371/ journal.pntd.0006382, ..). For researchers who are not familiar with sand flies (the targeted readers for this review), it is crucial to recognize that in field studies, the detection of parasites or their DNA in collected blood-fed females (before the defecation) is insufficient for vector incrimination.

11. On line 247 it is written that the function of haptomonad parasites is yet to be discovered. However, it is known for decades that haptomonads attach to the cuticular lining of the stomodeal valve (through an expanded flagellum containing hemidesmosomal structures) and damage it, facilitating reflux of parasites from the midgut (Schlein et al. 1992 doi.org/10.1073/pnas.89.20.9944). Surprisingly, the term stomodeal valve is not mentioned in the manuscript although its colonisation and damage is the prerequisite to successful *Leishmania* transmission (Volf et al 2004 doi.org/10.1016/j.ijpara.2004.07.010, Rogers et al 2008 <https://doi.org/10.1111/j.1462-5822.2008.01132.x>).

12. The effect of sand fly saliva on *Leishmania* infection should be more emphasized and described in more detail. At least, the four different phenomena should be mentioned – (I) protection of hosts by preimmunization with salivary glands lysates or their pre-exposure to uninfected sand fly bites (Belkaid et al 1998, <https://doi.org/10.1084/jem.188.10.1941>), (II) the enhancing effect of saliva co-inoculated with *leishmania* on disease progression (Titus and Ribeiro 1988, DOI: 10.1126/science.3344436), (III) the enhancing effect of repeated exposures of infected hosts to uninfected sand fly bites (Vojtkova et al 2021, doi.org/10.3389/fitd.2021.745104) and (IV), antigenic properties of salivary glands and potential use of anti-saliva antibodies as exposure markers (reviewed by Lestinova et al 2017 DOI 10.1371/journal.pntd.0005600).

13. Figure 4. Is this figure based on experimental data? It suggests that the transmission post a single blood meal is not possible. However, this is not true (see for example Ashwin et al 2021, doi.org/10.1038/s41467-020-20569-3). However, this is not true, transmission post a single blood meal has been achieved experimentally many times (see for example Ashwin et al 2021, doi.org/10.1038/s41467-020-20569-3) and it is also possible that the effect of multiple feedings is not a general rule but can be typical for some parasite-vector pairs only.

14. Conclusions, line 385: I cannot agree with the authors that they compiled all the information essential for new researchers. More appropriate would be „In this Review, we tried to compile information important for new researchers....“

Table 1.

- The list of proven vectors is not complete (compare to Maroli 2012), I suggest completing it or naming the slope as “main vectors”.
- For *L. infantum*, reservoir potential (infectivity for sand flies) has been demonstrated also for many other mammalian species: rats (Zanet et al. 2014, Gradoni et al. 1983, Pozio et al. 1985), rabbits (Jiménez et al. 2014), *Cercopithecus thomasi* (Gomes et al. 2007), *Didelphis albiventris* a *Didelphis marsupialis* (Sherlock 1996, Travi et al. 1994, Travi et al. 1998) cats (Maroli et al. 2007, da Silva et al. 2010), *Chrysocyon brachyurus* and *Speothos venaticus* (Mol et al. 2015) and non-human primates (De Oliveira et al. 2019).
- *L. siamensis* is nomen nudum, not a valid name, please change it for *L. orientalis*. It is present only in Thailand while *L. martiniquensis* is present in Martinique, Thailand, USA, Central Europe (Pothirat et al 2014, doi: 10.1371/journal.pntd.0003339, Jariyapan et al 2018, doi.org/10.1186/s13071-018-2908-3). In both *L. martiniquensis* and *L. siamensis*, the subgenus *Mundinia* should be written and for both, reservoirs and vectors are unknown while main clinical features have been published.
- *L. colombiensis* has been included in the genus *Endotrypanum* (Espinosa et al 2016, doi: 10.1017/S0031182016002092). It should be mentioned at least as a note.

Reviewer #2 (Remarks to the Author):

Reviewer #1:

I agree with the authors that there is a lack of information on sand flies and their interaction with

leishmania, and believe that any new and good quality information on this important and neglected vector is important. In this case the text is very well written, the reading flows well and easy. Although there are quite a few reviews related to this subject published over time, the present review focuses on some different aspects of the vector-parasite interaction, and is valid.

Comments:

I found Fig. 4 hard to understand; please try to make a little clearer.

Pages 191 and 194: I personally do not agree with the term escape, considering the PM eventually is degraded and the parasites are exposed in the gut.

285: with

294: not sure I understood how you can control one specific vector.

470: sand flies

Reviewer #3 (Remarks to the Author):

General commentary:

The work titled " Sand flies: Basic information on the vectors of leishmaniasis and their interactions with Leishmania parasites" it is interesting and addresses a subject very little discussed in relation to Leishmaniasis. The work develops aspects of the relationship between sandflies and Leishmania interaction, addressing the information about sand fly taxonomy, distribution, life cycle, blood-feeding process, development of mature infections, Leishmania transmission and outstanding questions. The authors present important work on vectors of leishmaniasis and their interactions with Leishmania parasites. Apart from minor grammatical errors especially tenses the manuscript is well presented.

I suggest the inclusion of a topic talking about the parasite's avoidance to the insect's response, which is even mentioned by the authors during the work.

I also suggest revising the grammatical errors of the paper. For example; Line 186, sustain mature infections, and transmit them to naïve hosts, causing disease.

In Abstract part: Line 30-31: "However, while many laboratories focus on the disease(s) and etiological agents, few study the respective vector. " In my opinion, there are many laboratories working on the sand fly. But I also agree that more work needs to be done on the vectors involved.

Line 103-104: "More than 900 sand fly species and subspecies are recognized and grouped within six genera" There are approximately 1,000 valid described species of sand flies in the world (Shimabukuro PHF, Andrade AJ, Galati EAB. Checklist of American sand flies (Diptera, Psychodidae, Phlebotominae): Genera, species, and their distribution. Zookeys 2017; 660: 67–106.)

Line 307-308: "With respect to the vector-derived infection enhancers, both the sand fly gut microbiota and sand fly saliva were demonstrated to play a role. "Literature should be included.

Point-by-point Rebuttal to the Reviewer's comments

Reviewer #1 -

This manuscript is focused on Phlebotomine sand flies, vectors of medically important Leishmania parasites. The author's aim was to "familiarize new researchers with these insects" - they compiled information on Leishmania – sand fly interactions with information on sand fly biology, ecology and behaviour. While the chapters focused on Leishmania – sand fly interactions are written in sufficient details (with exception of some, further pointed items), the parts describing sand fly taxonomy, biology and ecology are reduced on very basic facts. (...) Valuable is the last chapter summarizing the outstanding questions and pointing topics that should be answered to understand the complex interactions connected with leishmaniasis.

Response: First, we want to thank Reviewer #1 for the time spent reading and evaluating this manuscript. The authors appreciate the thorough revision and the many relevant comments that helped us to considerably improve our manuscript. We acknowledge most of the limitations pointed out, particularly with respect to taxonomy, biology, and ecology. On the other hand, we are happy that the Reviewer also found some strengths including the info on parasite-vector interactions and the outstanding questions chapter. Do note that, our aim is to "to familiarize ("naïve") researchers with sand flies, **in the context of leishmaniasis**". Therefore, this text is directed to all of those that lack elementary expertise in vector biology and entomology and are looking for basic information on sandflies. This said, we considered all of the issues raised by the Reviewer and revised the manuscript (yellow highlighted text), when applicable; please check below our point-by-point responses to your comments.

Specific comments:

0. It was, probably, the intention of authors but, in this case, I would expect references to recent papers or book chapters summarizing properly the biology of sand flies and their role in leishmaniasis transmission. To my surprise, references to two important recent reviews on sand fly biology, distribution and Leishmania transmission are missing (Maroli et al 2012, doi: 10.1111/j.1365-2915.2012.01034.x and Dvorak et al 2018,doi.org/10.1007/978-3-319-72386-0_1).

Response: Thank you for the remark. We apologize we did not include these references in this manuscript. In fact, they have helped a lot in the writing of the taxonomy, distribution, and life cycle sub-section. Now, in the Revised manuscript, both references are cited, together with a chapter in the same book by Prof. Luigi Gradoni on *Leishmania* epidemiology (https://doi.org/10.1007/978-3-319-72386-0_1). Please check the new references 21, 30, and 31.

1. According to the abstract, the authors should discuss the taxonomy of sand flies. However, this is not true; the taxonomy is provided on 5 rows only as a categorization of the subfamily Phlebotominae and a list of the 6 sand fly genera. This very simple introduction does not allow any deeper insight into a discussion on sand fly taxonomy. Authors should either change this sentence in the abstract or enrich the

text on sand fly taxonomy (Does the current taxonomy reflect the phylogeny of the group? What are the main morphological traits important for sand fly species discrimination? Which molecular methods have been used and how improved species identification and sand fly taxonomy? ...)

Response: Thank you for the comments. We agree with the Reviewer; considering the manuscript as it was, we should not have mentioned we performed an overview on the taxonomy of sandflies. In the revised manuscript, in line with the Reviewer's comment and the request from the Editorial office, we expanded on this subject. Do note that, because this Review is not directed to entomologists, and because that, as usual with respect to systematics and taxonomy, there is no consensus, we decided to do a brief overview, and direct the readers to relevant manuscripts. Given the above, we revised the title of the respective sub-section to "Taxonomy **at a glance, biology**, distribution, and life cycle" (line 100 of the revised manuscript). Additionally, we also deleted the mention of taxonomy from the Abstract, not to mislead potential readers that are specifically looking for detailed info on the taxonomy of phlebotomine sandflies. This said, we are also ready to delete the mention to taxonomy completely if the Reviewer thinks it would be more appropriate. The newly added text to the revised manuscript reads as follows:

"Sand flies are arthropods and insects included in the order Diptera (two-winged flies), suborder Nematocera, family Psychodidae, and subfamily Phlebotominae⁷. Around 1000 sand fly species/subspecies were validated/described thus far around the world²⁸. **Initially, the taxonomical classification of sandflies was based on morphological analyses, including first an external analysis also known as phlebotometry (e.g., observation of the male genitalia, and determination of the wing venation indices...), and then the investigation of internal structures such as the spermathecae, cibarium, and the pharynx^{7,29}. More recently, modern methods including chromosome analysis, isoenzyme analysis, molecular and phylogenetic analyses (DNA barcoding and Next-Generation Sequencing), and mass spectrometry (MALDI-TOF), allowed the better identification and classification of sandfly specimens and consequently the clarification of some variations within sandfly subgenera/populations^{7,30}. Many classification systems have been proposed over the years including those of Abonnenc, Davidson, Fairchild, Galati, Leng, Lewis, Quate, Rispaill & Légerand, Secombe, Theodor, and Young & Duncan^{7,28,30,31}. However, with respect to taxonomy and the classification of sandflies, there is still no consensus, reason why we decided not to describe each of the aforementioned classification systems in this Review; for more details, as well as a historical perspective on the taxonomy and systematics of sandflies, please check a few comprehensive Reviews on the subject^{7,28,30,31}. Instead, for the sake of simplicity, here we adopted the subdivision of the Phlebotominae into six genera, as per the widely accepted classification based on a conservative approach: *Phlebotomus* (13 subgenera), *Sergentomyia* (10 subgenera), and *Chinius* (four species) from the Old World, and *Lutzomyia* (26 subgenera and groups), *Brumptomyia* (24 species), and *Warileya* (six species) from the New World (Fig. 1)^{7,30}." (Lines 101-122 of the revised manuscript)**

2. Line 44: “more than 300 species feed on blood” the number is seriously underestimated; even bloodfeeding sand flies are almost 1000 species. Consider either changing it or deleting the entire sentence.

Response: Thank you for the remark. We apologize for this mistake. The sentence was revised for accuracy, as follows:

“Estimates point to the existence of 200 million insects alive per each human at any given point; among them, **around 14000 species** feed on blood¹, some, with potentially severe implications for human health.” (Lines 43-45 of the revised manuscript)

3. Line 76: The sentence „Evidence suggest that domestic dogs may be reservoirs of this parasite species “is supported with the citation of one manuscript (No.18) based on data from India. However, a role for animal reservoir hosts in the transmission of L. donovani has been suggested for many years in East Africa. Authors should mention this and cite at least two other sources: Elnaiem et al (2001) doi:10.1017/S0031182001007594, Dereure et al. (2003) doi.org/10.1016/j.micinf.2003.07.003s.

Response: Thank you for the remark. At this point, we are still not convinced that *L. donovani* is a zoonotic agent. The authors of the two mentioned articles were also very careful in the interpretation of their data and the incrimination of animals as the reservoirs of *L. donovani* – like us in this review they use words such as “potential”, “may be”. On the other hand, the authors of the study originally cited dare to incriminate dogs as *L. donovani* reservoirs, the reason why we chose it. However, we agree with the Reviewer that these articles are important for this particular context, and, consequently cited them in the revised manuscript. The sentence was updated accordingly, as follows:

“Only for *Leishmania donovani* parasites, no animals other than man have been incriminated as a reservoir (although evidence suggests that domestic dogs **and mongooses (*Herpestes ichneumon*)** may be reservoirs of this parasite species, **as per studies from India¹⁸ and East Africa^{19,20}**); for all of the remaining *Leishmania* species that cause disease in humans at least one animal reservoir (frequently sylvatic) is recognized (Table 1)^{8,21}.” (Lines 74-78 of the revised manuscript)

4. Lines 87-97: This part is very similar to Abstract (lines 31-38). Please, consider revision.

Response: Thank you for the suggestion. Such similarity was in fact intentional. While most readers only read a paper after reading the Abstract, a few jump straight to the Introduction after reading the Title. Therefore, to make sure that every reader understands the rationale behind this Review article, we mention it both in the Abstract and in the Introduction. In line with this, we maintained our rationale in both sections. However, in line with the Reviewer’s suggestion a few details were changed to make the text less *ipsis verbis*, as follows:

“Therefore, the complete understanding of leishmaniasis (as a “whole”), depends not only on the dissection of the clinical aspects (parasite-host interactions) but also on the comprehension of the sand fly vectors and their interactions with *Leishmania* parasites and the animal/human hosts. However, most laboratories around the globe focus exclusively on parasite-host interactions, disregarding the sand fly vectors. In fact, information in the literature on sand flies is not easily accessible, at least in a comprehensible fashion. Therefore, the familiarization of new researchers with the vectors of *Leishmania* parasites can be a challenging task. To address this issue, here, we compile the basic information on sand flies, **including the taxonomy (at a glance), biology, distribution, and life cycle, the blood-feeding process, and the *Leishmania*-sand fly interactions important for parasite transmission, as a resource for the scientific community in general.** Moreover, we also discuss the outstanding questions in the field, answers to which are essential for the complete understanding of the parasite-vector-host interactions that lead to leishmaniasis.” (Lines 85-96 of the revised manuscript)

5. The chapter about taxonomy, distribution and life cycle could be renamed “Biology, distribution and life cycle”. Two important reviews mentioned above should be referred.

Response: Thank you for the suggestion. The chapter was renamed for accuracy, in line with our answer to point 1, and your suggestion as “Taxonomy **at a glance, biology**, distribution, and life cycle” (line 100 of the revised manuscript). The two Reviews were also cited, as suggested.

6. Information about the life cycle and life tables on Lines 118-133: I am missing two valuable references on this topic, the first summarizing life table of about ten sand fly species under standard conditions (Volf and Volfova 2011, DOI 10.1111/j.1948-7134.2011.00106.x), and the second demonstrating the effect of temperature on sand fly life cycle (Benkova and Volf, 2007, DOI 10.1603/0022-2585).

Response: Thank you for the comment. Do note that the first article describes sandfly colonies under laboratory (ideal) conditions. Therefore, the timings in Nature may vary. This said, we cited this article, within the following sentence (new Reference 40):

“There are four larval instars, and larval development is usually completed in 20 to 30 days, depending on the sand fly species⁴⁰, as well as on the temperature and availability of food.” (Lines 138-140 of the revised manuscript)

The second Reference (new Reference 33) was also included to further support the following sentence:

“Still, humidity is an important factor that together with temperature, are detrimental for, **and influence** sand fly development³³⁻³⁵.” (Lines 128-129 of the revised manuscript)

7. The chapter dedicated to sand flies as vectors of *Leishmania* parasites should be introduced by the information that only 10% of described sand fly species are proven or suspected vectors of *Leishmania*. Authors should also provide criteria for vector competence specified by Killick-Kendrick (1990), doi.org/10.1111/j.1365-2915.1990.tb00255.x and WHO (2010) *Control of the leishmaniasis. Report of a Meeting of the WHO Expert Committee on the Control of Leishmaniases*.

Response: Thank you for this suggestion, which we believe is very pertinent. This info was added to the manuscript, accordingly, and the two references were cited (new References 55 and 56). The beginning of this section now reads:

“Among the 1000 sand fly species/subspecies validated/described thus far around the world²⁸, only one-tenth (10%) are proven or suspected vectors of *Leishmania* parasites^{31,54}. These meet all (proven vectors) or almost all of the vector incrimination criteria proposed by Killick-Kendrick⁵⁵ and the WHO Expert Committee on the control of Leishmaniases⁵⁶: i) they feed on humans (are anthropophilic), ii) they also feed on the relevant reservoir hosts in the case of zoonotic agents, iii) they are found in Nature infected with the same parasites (*Leishmania* species) circulating in humans (from the same geographical area); iv) they support the complete development of the *Leishmania* parasites circulating in humans, including after the defecation of the bloodmeal remnants; and v) they are able to transmit those parasites to susceptible hosts when they take a bloodmeal^{30,31}. Importantly, with respect to the sand fly vectors incriminated thus far, *Leishmania*-sand fly interactions studied under laboratory conditions led to **their separation into two major groups: restrictive and permissive vectors⁵⁷.”** (Lines 193-204 of the revised manuscript)

8. In the chapter “Sand fly – *Leishmania* interaction, authors should briefly describe the morphology of the sand fly digestive tract and define the foregut, midgut and hindgut and their functions before using these terms for an explanation of suprapylarian, peripylarian and hypopylarian development of parasites.

Response: Thank you for the remark. We followed your suggestion and briefly described the morphology of the sand fly digestive tract. The revised text reads as follows:

“Of note, contrary to many vector-borne agents (including some Trypanosomatids), the development of *Leishmania* parasites is confined to the sand fly digestive tract (there is no crossing/disruption of the epithelial barrier^{59,64}), simplistically divided here (excluding the crop) into: i) the foregut - the most anterior portion, from the mouth to the cardia, which includes the stomodeal valve; ii) the (thoracic and abdominal) midgut – from the cardia to the pylorus; and iii) the hindgut – the most posterior portion, from the pylorus to the rectum⁶⁵.” (Lines 234-239 of the revised manuscript)

9. On lines 214-220 authors cite old hypotheses that proteases pose the first and one of most significant barriers to parasite survival and procyclic promastigotes are more resistant than transforming stages. However, more recent experiments did not show any direct negative effect of proteases on transforming

parasites while documenting that leishmania mortality results from toxic products of blood meal digestion (Pruzinova et al 2018, DOI 10.1186/s13071-018-2613-2). Please, revise the text based on these findings.

Response: Thank you for the comment. We understand the Reviewer's point. However, there are more studies suggesting the importance of sand fly proteases on the establishment of infection by other *Leishmania* species, particularly from the New World. Therefore, we cited the mentioned paper in the revised manuscript but did not assume definitively that sandfly proteases and the differentiation of amastigotes into procyclic promastigotes are not important for the establishment of infection. Additionally, the toxicity of digestion byproducts was also mentioned in Fig. 3, for accuracy. The modified/added sentences read as follows:

“This first differentiation step **was proposed to be** extremely important; **it was postulated** that parasites within the blood meal need to resist digestive proteases, the first and one of the most significant barriers to parasite survival⁶⁷. **In one study, *L. major* procyclic promastigotes were demonstrated to be more resistant to proteolytic attack than the parasites in the transitional state (from amastigote to promastigote forms)⁶⁸; a possible explanation for such a phenotype is the known dynamic changes of the parasites' glycoalyx components: e.g. comparing amastigotes with promastigotes, the latter have a higher content of LPG in their membrane^{62,67}. On the other hand, a more recent study showed contrary findings, and the authors suggested parasite killing (*L. major* and *L. donovani*) could be due to the toxic products of blood meal digestion⁶⁹. However, although Pruzinova and colleagues suggest that proteases do not have a direct role in *Leishmania* mortality within the blood bolus, at this point such a generalization cannot be made, mainly because there are data from other vector-parasite pairings (*Lu. longipalpis* – *L. mexicana*/*L. infantum*) that may still suggest an implication of sand fly proteases in the establishment of *Leishmania* parasites within the vector⁷⁰⁻⁷². Whether such an effect is parasite-species specific and a direct consequence of sand fly proteases or an indirect one caused by toxic byproducts generated by the digestion of blood must still be clarified.” (Lines 245-260 of the revised manuscript)**

10. In this chapter, authors should also emphasize the important fact that the early phase of *Leishmania* development in the vector is nonspecific. Before defecation, almost any *Leishmania* species survive also in sand fly species, which are not competent vectors (Pruzinova et al 2018, Sadlova et al 2018 doi.org/10.1371/journal.pntd.0006382, ..). For researchers who are not familiar with sand flies (the targeted readers for this review), it is crucial to recognize that in field studies, the detection of parasites or their DNA in collected blood-fed females (before the defecation) is insufficient for vector incrimination.

Response: Thank you for the remark. We believe this is implicit since the parasites that do not attach to the midgut are defecated, as we mentioned in the original text. However, we also believe that the details about vector incrimination are well worth mentioning. Therefore, we added two sentences to the manuscript, in line with the Reviewer's suggestion, as follows:

“Of note, even in refractory vectors parasites can differentiate into promastigotes and multiply within the blood bolus, but are then eliminated via defecation^{69,74}; therefore, the “escape” from the peritrophic matrix and the attachment to the sand fly midgut to avoid the defecation-mediated elimination are detrimental for the establishment of *Leishmania* parasites within permissive sand flies. In line with this, it is important to repeat here one of the criteria that need to be met for the incrimination of sand flies as vectors of *Leishmania* parasites - sandflies support the complete development of the *Leishmania* parasites circulating in humans, including after the defecation of the bloodmeal remnants – and stress the fact that the detection of parasites or their DNA in engorged sand fly females (before the defecation) is insufficient for vector incrimination⁷⁶.” (Lines 274-283 of the revised manuscript)

11. On line 247 it is written that the function of haptomonad parasites is yet to be discovered. However, it is known for decades that haptomonads attach to the cuticular lining of the stomodeal valve (through an expanded flagellum containing hemidesmosomal structures) and damage it, facilitating reflux of parasites from the midgut (Schlein et al. 1992 doi.org/10.1073/pnas.89.20.9944). Surprisingly, the term stomodeal valve is not mentioned in the manuscript although its colonisation and damage is the prerequisite to successful *Leishmania* transmission (Volf et al 2004 doi.org/10.1016/j.ijpara.2004.07.010, Rogers et al 2008 <https://doi.org/10.1111/j.1462-5822.2008.01132.x>).

Response: Thank you for the remark. With respect to haptomonads we wanted to highlight two things: 1) their origin is unknown, and 2) they are the less studied (and thus the most “neglected”) vector-derived parasite form. Indeed, a few studies showed that the haptomonads attach to the cuticular lining of the stomodeal valve. However, we do not agree that are the haptomonads that damage the stomodeal valve, as suggested by the Reviewer; the papers mentioned (and others in the literature) do not allow such a conclusion (as they also do not allow the exclusion of such a hypothesis). The inner softer chitin layer of the stomodeal valve is degraded by chitinolytic enzymes produced by *Leishmania* parasites; however, whether these enzymes are produced by haptomonads or by other parasite forms within the plug is not clear, as far as we are concerned. Therefore, and because we admit that the original sentence was inaccurate, we revised it, as follows:

“Additionally, leptomonad promastigotes are also thought to attach to the sand fly stomodeal valve and give rise to haptomonad promastigotes, the less studied **(and thus the most “neglected”)** vector-derived parasite form, **whose role is not completely clear⁸⁰. This said, the attachment of these parasite forms may be important for the loss of the integrity of the stomodeal valve (together with the action of parasite-derived chitinolytic enzymes), which is important for the transmission process⁸¹⁻⁸³.”** (Lines 294-300 of the revised manuscript)

Moreover, we also mentioned the importance of the damage of the stomodeal valve for the transmission of *Leishmania* parasites (blocked fly hypothesis), as follows:

“Importantly, such a regurgitation is also facilitated by the *Leishmania*-induced damage of the sand fly stomodeal valve, known to be permanently opened in the context of heavy infections^{81,83}. Of note, as an indirect consequence of the formation of the PSG plug/damage of the stomodeal valve, the behavior of infected sand flies is also altered...” (Lines 340-344 of the revised manuscript)

12. The effect of sand fly saliva on *Leishmania* infection should be more emphasized and described in more detail. At least, the four different phenomena should be mentioned – (I) protection of hosts by preimmunization with salivary glands lysates or their pre-exposure to uninfected sand fly bites (Belkaid et al 1998, <https://doi.org/10.1084/jem.188.10.1941>), (II) the enhancing effect of saliva co-inoculated with *leishmania* on disease progression (Titus and Ribeiro 1988, DOI: 10.1126/science.3344436), (III) the enhancing effect of repeated exposures of infected hosts to uninfected sand fly bites (Vojtkova et al 2021, doi.org/10.3389/fitd.2021.745104) and (IV), antigenic properties of salivary glands and potential use of anti-saliva antibodies as exposure markers (reviewed by Lestinova et al 2017 DOI: 10.1371/journal.pntd.0005600).

Response: Thank you for the suggestion. In this review, we want to cover the most basic aspects pertaining to sand flies, and their interactions with *Leishmania* parasites. Thus, we purposely excluded the more translational aspects of sandfly research, including the mention of the host immune response to sandfly saliva and the potential of salivary proteins as anti-*Leishmania* vaccines and markers of exposure from this Review. Of note, these subjects were comprehensively reviewed many times in the past few years,– e.g. Abdeladhim, 2014 (<https://doi.org/10.1016/j.meegid.2014.07.028>), Lestinova, 2017 (<https://doi.org/10.1371/journal.pntd.0005600>), and Cecilio, 2018 (10.5772/intechopen.75000). Therefore, we just clarified that sand fly saliva, as part of the infectious inoculum, helps in the establishment of infection (Titus and Ribeiro, 1988), as well as exacerbates active disease (Vojtkova, 2021). The newly added text reads as follows:

“Of note, such an infection-enhancing effect of sand fly saliva was demonstrated *in vivo* both in the context of transmission (the establishment of infection)^{102,103}, and of active disease (cutaneous leishmaniasis mouse model)¹⁰⁴.” (Lines 377-379 of the revised manuscript)

Do note, that we briefly mentioned sand fly-derived anti-*Leishmania* vaccines in the outstanding questions, and now, we also mention the markers of exposure as disease control tools:

“Additionally, more and more sand fly-based vaccine candidates against leishmaniasis^{116,117} are being proposed as essential disease control tools that can be used in combination with the *Leishmania*-derived ones^{118,119}. The same is true considering the exploitation of sand fly salivary proteins as markers of exposure, as tools for the control of leishmaniasis^{46,95}.” (Lines 430-434 of the revised manuscript)

13. Figure 4. Is this figure based on experimental data? It suggests that the transmission post a single blood meal is not possible. However, this is not true, transmission post a single blood meal has been achieved experimentally many times (see for example Ashwin et al 2021, doi.org/10.1038/s41467-020-20569-3) and it is also possible that the effect of multiple feedings is not a general rule but can be typical for some parasite-vector pairs only.

Response: The figure was based on the publication that multiple blood meals increase vector competence (Serafim et al. 2018 - <https://doi.org/10.1038/s41564-018-0125-7>). We are actually convinced that transmission after a single blood meal is the exception and not the rule. This said, we agree with the Reviewer that transmission post a single blood meal may be also occurring in nature and can clearly be set up experimentally.

In the paper the Reviewer mentions (a very nice paper by the way), as in most of the papers dealing with *Leishmania* transmission by sandflies, the system is pushed to the limit to favor infection and transmission:

1. The sandflies are fed with a high concentration of parasites [in fact, the infection rates are dependent on the initial concentration of parasites in the bloodmeal, as reported in many exploratory papers – e.g. Myskova et al. 2008 (<https://doi.org/10.1093/jmedent/45.1.133>)];
2. Only the fully engorged flies are used;
3. Transmission (or pick up) is carried out with multiple sandflies per vial (10-20) in the context of anesthetized animals.

Consequently, and although these studies are invaluable for the advancement of the knowledge on sandflies, a parallel to the natural context cannot be drawn:

1. Even xenodiagnosis in the context of symptomatic dogs (*Lu. longipalpis* – *L. infantum*) - **anesthetized** - (as an example) only led to the infection of less than 30 % of the sandflies (with an infectivity of 83%), a number that decreased to 5 % in oligosymptomatic and asymptomatic dogs (Michalsky et al. 2007 - <https://doi.org/10.1016/j.vetpar.2007.03.004>). The same was true in the context of the *P. perniciosus* - *L. infantum* pairing; sandflies fed on non-restrained **relaxed** dogs (given Adaptil® Express Calming tablets) showed infection rates ranging from 0 to 79% with a maximum potential transmissibility rate of 64% (Gizzarelli et al. 2021 - <https://doi.org/10.3389/fvets.2021.667290>). **This suggests that even in the context of highly transmissible reservoirs and fully engorged flies the final infection rates and the quality of infection are not spectacular – opposing the artificial infections optimized for the achievement of high numbers of parasites and high frequencies of metacyclics/gut.**
2. Another layer that must be considered is the feeding efficiency. While under laboratory conditions the host interference is eliminated or limited (e.g. use of anesthetized animals, which is the rule), in the field the feeding will probably be interrupted. **Therefore, at least some sandflies are expected not to fully engorge, or to achieve that via re-feeding attempts in the same or multiple hosts (that will not all be infected with *Leishmania* parasites).** Consequently, the infection/transmissibility rates mentioned above with dogs, are expected to be even lower.
3. Additionally, although the determinants of transmission in the field are yet to be fully understood, we believe it is not very much likely that one individual will be exposed to the bites of at least 5 heavily infected sandflies at once in exactly the same area of the body – something done in the experimental models. Do note that we do not know if a single heavily infected

sandfly can transmit e.g. visceral disease to humans, as stated in the outstanding questions, although we do know that bites of single flies after a second bloodmeal lead to a significantly higher chance of development of lesions (*versus* flies given a single bloodmeal) in a mouse model of cutaneous leishmaniasis (*P. papatasi* – *L. major* pairing; Serafim et al. 2018 - <https://doi.org/10.1038/s41564-018-0125-7>).

Considering the above, we are convinced that in nature, via the acquisition of multiple bloodmeals, even sandflies that initially did not pick up high numbers of parasites, have the potential to become heavily infected and transmit the disease. This is what we ultimately want to convey with the Figure. However, since we cannot exclude the possibility that sandflies can be infectious in the field after a single blood meal (assuming they will not seek a subsequent bloodmeal 6 days after the first – which we do not know if it is likely), we revised Figure 4 and its legend, for accuracy, as follows:

“Fig. 4. The impact of multiple blood-meals on the maturation of *Leishmania* infections within the sand fly vector. In nature, sand flies are expected to feed on blood multiple times for the completion of more than one gonotrophic cycle. Importantly, the intake of multiple bloodmeals (represented by the red blood drops) is expected to impact the vector competence, promoting not only the increase in the absolute parasite numbers (yellow) but also in both the percentage and number of the metacyclic infectious forms (green) in the sand fly midgut. **Importantly, such an increase in total parasite numbers (A), and particularly in the number of metacyclic promastigotes (B) in the midgut of infected sandflies that take subsequent blood meals, compared with single-fed flies (blue lines), results in a higher probability of transmission of *Leishmania* parasites (purple gradient).** Courtesy NIAID.” (Lines 908-919 of the revised manuscript)

14. Conclusions, line 385: I cannot agree with the authors that they compiled all the information essential for new researchers. More appropriate would be „In this Review, we tried to compile information important for new researchers....“

Response: Thank you for the remark. As pointed out, we toned down the statement, as follows:

“In this review, **we tried to compile information**, in our opinion, essential for new researchers to become familiarized with sand flies, in the context of leishmaniasis.” (Lines 456-458 of the revised manuscript)

Table 1.

• The list of proven vectors is not complete (compare to Maroli 2012), I suggest completing it or naming the slope as “main vectors”.

Response: Thank you for the remark. The Reviewer is correct; the list is not complete. However, our intention was to mention the main vectors (with reproducible and independent incrimination data), which is not accurately conveyed by the Slope naming. Therefore, for the sake of accuracy, we re-named the slope, as suggested. Additionally, we added a note highlighting two reviews where the readership can find lists including more “vectors” including the suspected ones (Akhoundi et al. 2018 and Maroli et al. 2012).

• For *L. infantum*, reservoir potential (infectivity for sand flies) has been demonstrated also for many other mammalian species: rats (Zanet et al. 2014, Gradoni et al. 1983, Pozio et al. 1985), rabbits (Jiménez et al. 2014), *Cerdocyon thous* (Gomes et al. 2007), *Didelphis albiventris* a *Didelphis marsupialis* (Sherlock 1996, Travi et al. 1994, Travi et al. 1998) cats (Maroli et al. 2007, da Silva et al. 2010), *Chrysocyon brachyurus* and *Speothos venaticus* (Mol et al. 2015) and non-human primates (De Oliveira et al. 2019).

Response: Thank you for the remarks. Here, we want to mention the main reservoirs (as the slope indicates); those that were undoubtedly proven to be involved in the transmission of parasites to humans. Although we acknowledge that the abovementioned animals can be infected with *L. infantum* and transmit the parasites (in some cases) to sandflies, we believe that for some, we still cannot exclude the hypothesis that, as what happens in humans, they are accidental hosts. Therefore, we updated the list with only a few of the mentioned “potential reservoirs” - rats, rabbits, foxes, and opossums.

• *L. siamensis* is nomen nudum, not a valid name, please change it for *L. orientalis*. It is present only in Thailand while *L. martiniquensis* is present in Martinique, Thailand, USA, Central Europe (Pothirat et al 2014, doi:10.1371/journal.pntd.0003339, Jariyapan et al 2018, doi.org/10.1186/s13071-018-2908-3). In both *L. martiniquensis* and *L. siamensis*, the subgenus *Mundinia* should be written and for both, reservoirs and vectors are unknown while main clinical features have been published.

Response: Thank you for the remarks. We updated the Table, for accuracy, as highlighted in yellow; the two mentioned references were also cited, together with a different one – Desbois et al. 2014 (10.1051/parasite/2014011). Do note that we kept the clinical features as unknown since there are not many cases, and thus we believe that we cannot yet distinguish typical from atypical cases.

• *L. colombiensis* has been included in the genus *Endotrypanum* (Espinosa et al 2016, doi:10.1017/S0031182016002092). It should be mentioned at least as a note.

Response: Thank you for the remark. We originally omitted this fact, but now it is highlighted in the form of a note.

Reviewer #2 -

I agree with the authors that there is a lack of information on sand flies and their interaction with leishmania, and believe that any new and good quality information on this important and neglected vector is important. In this case the text is very well written, the reading flows well and easy. Although there are quite a few reviews related to this subject published over time, the present review focuses on some different aspects of the vector-parasite interaction, and is valid.

Response: First, we want to thank Reviewer 2 for the time spent reading and evaluating this manuscript. The authors are very grateful for the positive and constructive evaluation of our work. We considered all the comments and revised the manuscript (yellow highlighted text), when applicable; please check below our point-by-point responses to your comments.

I found Fig. 4 hard to understand; please try to make a little clearer.

Response: Thank you for the remark. In line with your comment, we did slight changes to the figure, and to the legend, to allow an easier interpretation, as follows:

“Fig. 4. The impact of multiple blood-meals on the maturation of *Leishmania* infections within the sand fly vector. In nature, sand flies are expected to feed on blood multiple times for the completion of more than one gonotrophic cycle. Importantly, the intake of multiple bloodmeals (represented by the red blood drops) is expected to impact the vector competence, promoting not only the increase in the absolute parasite numbers (yellow) but also in both the percentage and number of the metacyclic infectious forms (green) in the sand fly midgut. **Importantly, such an increase in total parasite numbers (A), and particularly in the number of metacyclic promastigotes (B) in the midgut of infected sandflies that take subsequent blood meals, compared with single-fed flies (blue lines), results in a higher probability of transmission of *Leishmania* parasites (purple gradient).** Courtesy NIAID.” (Lines 908-919 of the revised manuscript)

Pages 191 and 194: I personally do not agree with the term escape, considering the PM eventually is degraded and the parasites are exposed in the gut.

Response: Thank you for the remark. We used this term, precisely because of such degradation that, according to some studies is mediated in part by parasite-derived components (e.g. chitinases) - Ramalho-Ortigao and Traub-Cseko 2003 (doi: 10.1016/s0965-1748(02)00209-6); Ramalho-Ortigao et

al. 2005 (doi: 10.1111/j.1365-2583.2005.00601.x). Additionally, there are reports suggesting that in unnatural parasite-vector combinations, the PM does not break down and that parasites are excreted with the bloodmeal remnants – e.g. Walters et al. 1992 (doi: 10.4269/ajtmh.1992.46.211), Sadlova et al. 2018 (<https://doi.org/10.1371/journal.pntd.0006382>). However, we do acknowledge other studies that suggest that such components (particularly the chitinases) may not have an important role in the “escape” of *Leishmania* parasites from the PM – e.g. Sádlová & Volf, 2009 (doi: 10.1007/s00441-009-0802-1). We cannot exclude the hypothesis that for some *Leishmania* species such degradation is relevant while for others it is not. Therefore, considering all of the above, together with the Reviewer’s suggestion, in the revised manuscript we used the term escape with quotations marks (please check lines 267, 270, 276, and 901 of the revised manuscript).

285: with

Response: We apologize for the typo in a previous version of our manuscript. This was revised accordingly (line 398 of the revised manuscript). Additionally, we made sure that no other similar typo was present in the revised manuscript.

294: not sure I understood how you can control one specific vector.

Response: As we clarify in the manuscript, this pertains to a speculative exercise. We are assuming that the vectors are not ubiquitous, meaning that within the Mediterranean Basin, some are probably prevalent in some regions, while others are prevalent in different regions. Therefore, what we want to highlight here is that if control strategies focus on some species – single or multiple – but not all permissive species (note that the control teams may not be aware of all of the species that are permissive to *L. infantum* parasites), the ones that remain may take the place as main vectors. Because we previously used the parenthesis “or multiple, but not all” in this section, we believe we were not implying that the control of a single species would be possible. However, because we admit the sentence may convey that idea, we revised it for accuracy, as follows:

“As a speculative exercise, thinking on the Mediterranean Basin, where at least eight different sand fly species were incriminated as vectors of *L. infantum* parasites⁹⁴, **to focus the control strategy on the main vector species in the area, but not in all permissive sand fly species** may have little/limited impact on disease control, considering the possibility of redundancy.” (Lines 407-411 of the revised manuscript)

470: sand flies

Response: We corrected this typo as well, detected in a previous version of our manuscript, accordingly (line 880 of the revised manuscript).

Reviewer #3 -

General commentary:

The work titled " Sand flies: Basic information on the vectors of leishmaniasis and their interactions with Leishmania parasites" it is interesting and addresses a subject very little discussed in relation to Leishmaniasis. The work develops aspects of the relationship between sandflies and Leishmania interaction, addressing the information about sand fly taxonomy, distribution, life cycle, blood-feeding process, development of mature infections, Leishmania transmission and outstanding questions. The authors present important work on vectors of leishmaniasis and their interactions with Leishmania parasites. Apart from minor grammatical errors especially tenses the manuscript is well presented.

Response: We want to thank Reviewer #3 for the time spent reading and evaluating this manuscript. The authors appreciate the positive evaluation, as well as the feedback and comments on our work. We considered all the comments and revised the manuscript (yellow highlighted text), when applicable; please check below our point-by-point responses to your comments.

I suggest the inclusion of a topic talking about the parasite's avoidance to the insect's response, which is even mentioned by the authors during the work.

Response: If we understood well, the Reviewer is asking us to discuss whether parasites subsist within sand flies because they escape the sand fly immune system. With respect to vector immunity, we are very interested in the topic "immune priming", more and more talked about. However, the information available in the literature mainly pertains to mosquitoes and *Plasmodium* or viral infection. Importantly, the life cycle of such pathogens within mosquitoes is quite different than the one of *Leishmania* parasites within sandflies; one of the differences that should be considered with this respect is the location of the infection – restricted to the midgut in the case of *Leishmania* with no breach of the epithelial barrier Vs. "systemic infection" in the case of *Plasmodium* and viruses, with the breach of the intestinal barrier. Importantly, the breach of the intestinal barrier seems to be important for the initiation of "harsher" immune responses, at least in mosquitoes, as well as to the development of immune priming, mediated by the gut microbiota - Crompton et al. 2014 (10.1146/annurev-immunol-032713-120220).

This said, we feel that pertaining to sand fly immunity, including the occurrence of immune priming, we know very little. Importantly, what is known was discussed in a brilliant Review published recently (2018) by Telleria et al. (<https://doi.org/10.1017/S0031182018001014>). Therefore, instead of covering this topic in detail, we added a paragraph to the "outstanding questions section" on sand fly immunity and *Leishmania*, and cited the abovementioned Review as good support for the readers who want to know more about the subject. The new paragraph, written to address your comment, reads as follows:

"Additionally, also on the topic of vector control, it is of paramount importance to disclose the immune responses in sand flies, in the context of *Leishmania* infection. And although more than a few breakthroughs were made on this topic in the last years, as recently reviewed in detail by Telleria and colleagues⁹⁶, much is yet unknown, particularly with respect to the existence of immune-related determinants of vector refractoriness. The fact that both the modulation of the gut microbiota^{97,111,112} and infection with particular viral agents¹¹² impact the establishment of *Leishmania* parasites within the sand fly vector, makes us wonder whether more than the simple competition, this is due to some

kind of “immune priming”, as reported for other relevant vectors of human disease^{114,115}; this hypothesis is worth to be explored in the future. Importantly, only when we comprehensively understand the sand fly immune responses detrimental for the elimination of *Leishmania* parasites can we start trying to answer the question: can we modulate immunity in sand flies to make them refractory to *Leishmania* parasites and this as a vector-based strategy for the control of leishmaniasis?” (Lines 414-426 of the revised manuscript)

I also suggest revising the grammatical errors of the paper. For example; Line 186, sustain mature infections, and transmit them to naïve hosts, causing disease.

Response: Thank you for the remark. Although we could not detect the grammatical inconsistency highlighted, we agree with the Reviewer that this sentence could read better – it was rather long. Therefore, the text was revised for the sake of readability, as follows:

“For instance, recently, we **reported** that *L. longipalpis* sand flies, vectors of *Leishmania infantum* parasites in nature, are competent vectors of *Leishmania major* parasites under laboratory conditions. **We demonstrated** that *L. longipalpis* sand flies are able to acquire *L. major* parasites from cutaneous leishmaniasis active lesions, to sustain mature infections, **and to transmit the parasites** to naïve hosts causing disease⁵⁵.” (Lines 207-212 of the revised manuscript)

Additionally, we re-read the whole manuscript and corrected any grammatical errors that we found.

In Abstract part: Line 30-31: "However, while many laboratories focus on the disease(s) and etiological agents, few study the respective vector. " In my opinion, there are many laboratories working on the sand fly. But I also agree that more work needs to be done on the vectors involved.

Response: Thank you for the remark. What we want to convey with this sentence is that the proportion of laboratories that study sand flies (or include sandflies in their studies) is considerably lower than that of laboratories that study *Leishmania* parasites and Leishmaniasis. Therefore, in line with your comment, and for accuracy, we revised this sentence, as follows:

“However, while many laboratories focus on the disease(s) and etiological agents, **considerably less** study the respective vectors.” (Lines 30-32 of the revised manuscript)

Line 103-104: "More than 900 sand fly species and subspecies are recognized and grouped within six genera" There are approximately 1,000 valid described species of sand flies in the world (Shimabukuro PHF, Andrade AJ, Galati EAB. Checklist of American sand flies (Diptera, Psychodidae, Phlebotominae): Genera, species, and their distribution. Zookeys 2017; 660: 67–106.)

Response: Thank you for the remark. For the sake of accuracy, we revised the sentence and cited the suggested reference in the revised manuscript, as follows:

“Around 1000 sand fly species/subspecies were validated/described thus far around the world²⁸.”
(Lines 102-103 of the revised manuscript)

Do note that, as suggested by Reviewer 1, we added a few more details on sand fly taxonomy, keeping, however, the info relatively simple not to defer the readership from reading the full paper. Please check the added details on lines 103-122 of the revised manuscript.

Line 307-308: “With respect to the vector-derived infection enhancers, both the sand fly gut microbiota and sand fly saliva were demonstrated to play a role. “Literature should be included.

Response: Thank you for this remark. We did not cite any references to support this sentence in the original manuscript since this is an introductory sentence. The two subjects are discussed separately in the subsequent sentences and the relevant references are cited. However, in line with the Reviewer’s suggestion, we cited a few comprehensive reviews on the subjects (new References 94-96):

94. Andrade, B.B., De Oliveira, C.I., Brodskyn, C.I., Barral, A. and Barral-Netto, M. (2007), Role of Sand Fly Saliva in Human and Experimental Leishmaniasis: Current Insights. *Scandinavian Journal of Immunology*, 66: 122-127. <https://doi.org/10.1111/j.1365-3083.2007.01964.x>.
95. Lestinova T, Rohousova I, Sima M, de Oliveira CI, Volf P (2017) Insights into the sand fly saliva: Blood-feeding and immune interactions between sand flies, hosts, and *Leishmania*. *PLOS Neglected Tropical Diseases* 11(7): e0005600. <https://doi.org/10.1371/journal.pntd.0005600>.
96. Telleria, E., Martins-da-Silva, A., Tempone, A., & Traub-Csekö, Y. (2018). Leishmania, microbiota and sand fly immunity. *Parasitology*, 145(10), 1336-1353. doi:10.1017/S0031182018001014.

If there is any particular Reference that is relevant for the subject and we forgot to mention, please let us know and we will gladly cite it.

REVIEWERS' COMMENTS:

Reviewer #1 (Remarks to the Author):

I appreciate the effort authors made on the manuscript; they considered all the comments and revised carefully the text. I recommend it for publication in *Communications Biology* with only one minor reservation.

The following point (the text focused on the effect of sand fly proteases on *Leishmania* infection, Point 9, lines 254-260) should be more clarified in the text:

There is no contradiction between the results of Pruzinova et al 2018 (*Leishmania* are destructed by products of blood meal digestion but not directly by midgut proteases) and the results of several studies showing that *Leishmania* may profit from reduced or delayed midgut proteolytic activities (Borovsky and Schlein 1987, *Med Vet Entomol.* 1987; 1:235–42; Schlein and Jacobson 1998 *Parasitology*; Dillon and Lane 1993 *Parasitol Res.* 79:492–6; Sant 'Anna et al 2009 *Parasites and Vectors*; Fernandes et al 2020 *Insect Biochemistry and Molecular Biology*), as reduced or delayed proteolysis prolongs the parasites hospitable environment without toxic products of blood meal digestion. I suggest modifying the text in this respect.

Otherwise, I feel only two more points should be commented on:

1) Ad point 11 – the role of haptomonads (lines 294-300)

I agree with the authors that haptomonads are the less studied *Leishmania* forms. Their mechanical role in the colonization of the stomodeal valve is well known but their involvement in chitinase production is not sure, so the current modification of the text is optimal.

2) Ad point 13 - Figure 4

I appreciate that authors adapted the Figure 4.

Transmission parameters of *Leishmania* parasites are an interesting point to discuss. Authors are true that in experimental infections, sand flies may engorge relatively high doses of parasites. On the other hand, parasite cultures used for experimental infections are often long-term maintained which may decrease their virulence in comparison with parasites circulating in nature.

What is the “natural dose” taken by sand flies on the infected host is an important transmission parameter which is still understood. For sure, *Leishmania* are distributed heterogeneously in the host skin (e.g. Doehl et al. 2017, *Nature Communications*) and, therefore, sand flies feeding on the same host take highly different numbers of amastigotes (as is apparent from xenodiagnostic studies mentioned also in the authors' answer). Huge infections may develop in females feeding on the “right” side while in most females, no parasites are taken or only their low numbers and in this case, a second bloodmeal may be necessary to enhance transmission. Multiple bloodmeals may be very helpful but transmission after the single bloodmeal also occurs and current data are insufficient to put a clear effective transmission threshold.

For these reasons, I am convinced that the modified Figure 4 better fits our current knowledge of the natural infection pattern.

Point-by-point Rebuttal to the Reviewer's comments

Reviewer #1 -

I appreciate the effort authors made on the manuscript; they considered all the comments and revised carefully the text. I recommend it for publication in Communications Biology with only one minor reservation.

Response: We would like to take the opportunity to thank again Reviewer #1 for the time spent reading and evaluating this manuscript, and for the detailed constructive feedback. We are convinced that this revised version has much improved, benefiting for the Reviewer's suggestions that could only be done by an expert on sandflies. We are also glad to know that our previous revisions were overall satisfactory, excluding the mentioned reservation, which we address below.

The following point (the text focused on the effect of sand fly proteases on Leishmania infection, Point 9, lines 254-260) should be more clarified in the text:

There is no contradiction between the results of Pruzinova et al 2018 (leishmania are destructed by products of blood meal digestion but not directly by midgut proteases) and the results of several studies showing that leishmania may profit from reduced or delayed midgut proteolytic activities (Borovsky and Schlein 1987, Med Vet Entomol. 1987;1:235–42; Schlein and Jacobson 1998 Parasitology; Dillon and Lane 1993 Parasitol Res. 79:492–6; Sant 'Anna et al 2009 Parasites and Vectors; Fernandes et al 2020 Insect Biochemistry and Molecular Biology), as reduced or delayed proteolysis prolongs the parasites hospitable environment without toxic products of blood meal digestion. I suggest modifying the text in this respect.

Response: Thank you for the comments. After reading the text once more, we agree with the Reviewer that the studies may not be contradictory. Therefore, for the sake of accuracy, we simplified the message, mentioning that either directly, or indirectly, proteases may impact the establishment of *Leishmania* parasites within the sand fly midgut (not excluding any of the hypotheses). The respective text in the revised manuscript now reads as:

“This first differentiation step was proposed to be extremely important; it was postulated that parasites within the blood meal need to resist **the effect of** digestive proteases, the first and one of the most significant barriers to parasite survival⁶⁷. In one study, *L. major* procyclic promastigotes were demonstrated to be more resistant to proteolytic attack than the parasites in the transitional state (from amastigote to promastigote forms)⁶⁸; a possible explanation for such a phenotype is the known dynamic changes of the parasites' glycocalyx components: e.g. comparing amastigotes with promastigotes, the latter have a higher content of LPG in their membrane^{62,67}. On the other hand, a more recent study showed contrary findings, and the authors suggested parasite killing (*L. major* and *L. donovani*) could be due to the toxic products of blood meal digestion⁶⁹. **Importantly, although contradictory, these studies both suggest the impact of midgut proteases in the establishment of *Leishmania* parasites within the vector (either directly or indirectly); this notion is further supported by studies focusing on other**

vector-parasite pairings (*Lu. longipalpis* – *L. mexicana/L. infantum*) showing that *Leishmania* parasites thrive after the downregulation of the proteolytic activity in the sand fly midgut⁷⁰⁻⁷².” (Lines 242-256 of the revised manuscript)

Otherwise, I feel only two more points should be commented on:

1) Ad point 11 – the role of haptomonads (lines 294-300)

I agree with the authors that haptomonads are the less studied leishmania forms. Their mechanical role in the colonization of the stomodeal valve is well known but their involvement in chitinase production is not sure, so the current modification of the text is optimal.

Response: Thank you for the remark. Once again, we thank you for the previous feedback that helped us highlight the neglected haptomonads in an accurate fashion.

2) Ad point 13 - Figure 4

I appreciate that authors adapted the Figure 4.

Transmission parameters of *Leishmania* parasites are an interesting point to discuss. Authors are true that in experimental infections, sand flies may engorge relatively high doses of parasites. On the other hand, parasite cultures used for experimental infections are often long-term maintained which may decrease their virulence in comparison with parasites circulating in nature.

What is the “natural dose” taken by sand flies on the infected host is an important transmission parameter which is still understood. For sure, *Leishmania* are distributed heterogeneously in the host skin (e.g. Doehl et al. 2017, Nature Communications) and, therefore, sand flies feeding on the same host take highly different numbers of amastigotes (as is apparent from xenodiagnostic studies mentioned also in the authors' answer). Huge infections may develop in females feeding on the “right” side while in most females, no parasites are taken or only their low numbers and in this case, a second bloodmeal may be necessary to enhance transmission. Multiple bloodmeals may be very helpful but transmission after the single bloodmeal also occurs and current data are insufficient to put a clear effective transmission threshold.

For these reasons, I am convinced that the modified Figure 4 better fits our current knowledge of the natural infection pattern.

Response: We are glad that the Reviewer was happy with the modifications we did to Figure 4. Indeed, the laboratory versus ecological dichotomy is quite interesting. Although the results obtained in the laboratory are invaluable, their translation to the natural context cannot be made in a straightforward fashion. Therefore, as we hope we stress in the outstanding questions, there is still a lot to learn with respect to sandflies, particularly considering the field. We thank the Reviewer one last time for the previous feedback, who helped us to generate a more accurate Figure.